# SNARE mimicry by the CD225 domain of IFITM3 enables regulation of homotypic late endosome fusion

Kazi Rahman [ID] [1,2], Isaiah Wilt[1], Abigail A Jolley[1], Bhabadeb Chowdhury[1], Siddhartha A K Datta[1] & Alex A Compton [ID] [1✉]

## Abstract

The CD225/Dispanin superfamily contains membrane proteins that regulate vesicular transport and membrane fusion events required for neurotransmission, glucose transport, and antiviral immunity. However, how the CD225 domain controls membrane trafficking has remained unknown. Here we show that the CD225 domain contains a SNARE-like motif that enables interaction with cellular SNARE fusogens. Proline-rich transmembrane protein 2 (PRRT2) encodes a SNARE-like motif that enables interaction with neuronal SNARE proteins; mutations in this region disrupt SNARE binding and are linked to neurological disease. Another CD225 member, interferon-induced transmembrane protein 3 (IFITM3), protects cells against influenza A virus infection. IFITM3 interacts with SNARE proteins that mediate late endosome-late endosome (homotypic) fusion and late endosome-lysosome (heterotypic) fusion. IFITM3 binds to syntaxin 7 (STX7) in cells and in vitro, and mutations that abrogate STX7 binding cause loss of antiviral activity against influenza A virus. Mechanistically, IFITM3 disrupts assembly of the SNARE complex controlling homotypic fusion and accelerates the trafficking of endosomal cargo to lysosomes. Our results suggest that SNARE modulation plays a previously unrecognized role in the diverse functions performed by CD225 proteins.

**Keywords** Membrane Fusion; Virus; IFITM; PRRT2; CD225
**Subject Categories** Membranes & Trafficking; Microbiology, Virology & Host Pathogen Interaction

## Introduction

Membrane fusion is a critical process enabling cargo transfer between membrane-bound compartments. The soluble NSF attachment protein receptor (SNARE) proteins are key mediators of membrane fusion and are involved in nearly every vesicular trafficking event in eukaryotic cells (Hong, 2005; Jahn and Scheller, 2006). SNARE proteins contain a so-called SNARE motif consisting of heptad repeats of generally hydrophobic amino acids surrounding a central hydrophilic residue (R for arginine or Q for glutamine) (Fasshauer et al, 1998). The central residue is referred to as the "0" layer while the hydrophobic residues surrounding it are numbered relative to the "0" layer, from "−7" to "+8" (Fig. 1A). These features enable the assembly of four parallel alpha-helices (one R-SNARE motif and three Q-SNARE motifs) into a coiled-coil trans-SNARE complex across two membranes. Progressive folding of the bundle towards a cis-SNARE complex, in which the SNARE proteins reside within the same membrane, drives membrane fusion (Han et al, 2017). Humans encode a minimum of 38 SNARE proteins which interact selectively to impart specificity and directionality to membrane fusion events between different organelles (Kloepper et al, 2007).

Among the best studied are the synaptic SNARE proteins that mediate the exocytosis of neurotransmitters from neurons (Sudhof, 2004). Neurotransmitter release at neuronal synapses requires the fusion of synaptic vesicles with the presynaptic plasma membrane, and this requires the assembly of syntaxin 1A and SNAP-25 (in the presynaptic plasma membrane) with VAMP2 (in the synaptic vesicle membrane) (Sutton et al, 1998; Xiao et al, 2001). The study of neurological diseases associated with impaired neurotransmission has led to the identification of cellular proteins that regulate synaptic SNARE assembly, such as proline-rich transmembrane protein 2 (PRRT2) (Ebrahimi-Fakhari et al, 2015). Mutations in PRRT2 associated with a range of neurological disorders in humans (including paraoxysmal kinesigenic dyskinia (PKD), infantile epilepsy and convulsions, hemiplagic migraines, ataxia, and intellectual disability) indicated that PRRT2 plays important roles in neurotransmission (Ebrahimi-Fakhari et al, 2015). Subsequently, it was shown that PRRT2 interacts with Q-SNAREs syntaxin 1A and SNAP-25 and disfavors their interaction with the R-SNARE VAMP2, thereby reducing synaptic SNARE assembly and neurotransmitter release (Coleman et al, 2018; Ma et al, 2018; Tan et al, 2018; Valente et al, 2016). However, a comprehensive understanding of the protein–protein interface enabling this regulatory function of PRRT2 was unknown. PRRT2 belongs to the CD225 superfamily, a group that includes trafficking regulator of GLUT4 (TRARG1/TUSC5) and interferon-induced

---

[1]HIV Dynamics and Replication Program, Center for Cancer Research, National Cancer Institute, Frederick, MD, USA. [2]Present address: Department of Biochemistry and Microbiology, School of Health and Life Sciences, North South University, Dhaka, Bangladesh. ✉E-mail: alex.compton@nih.gov

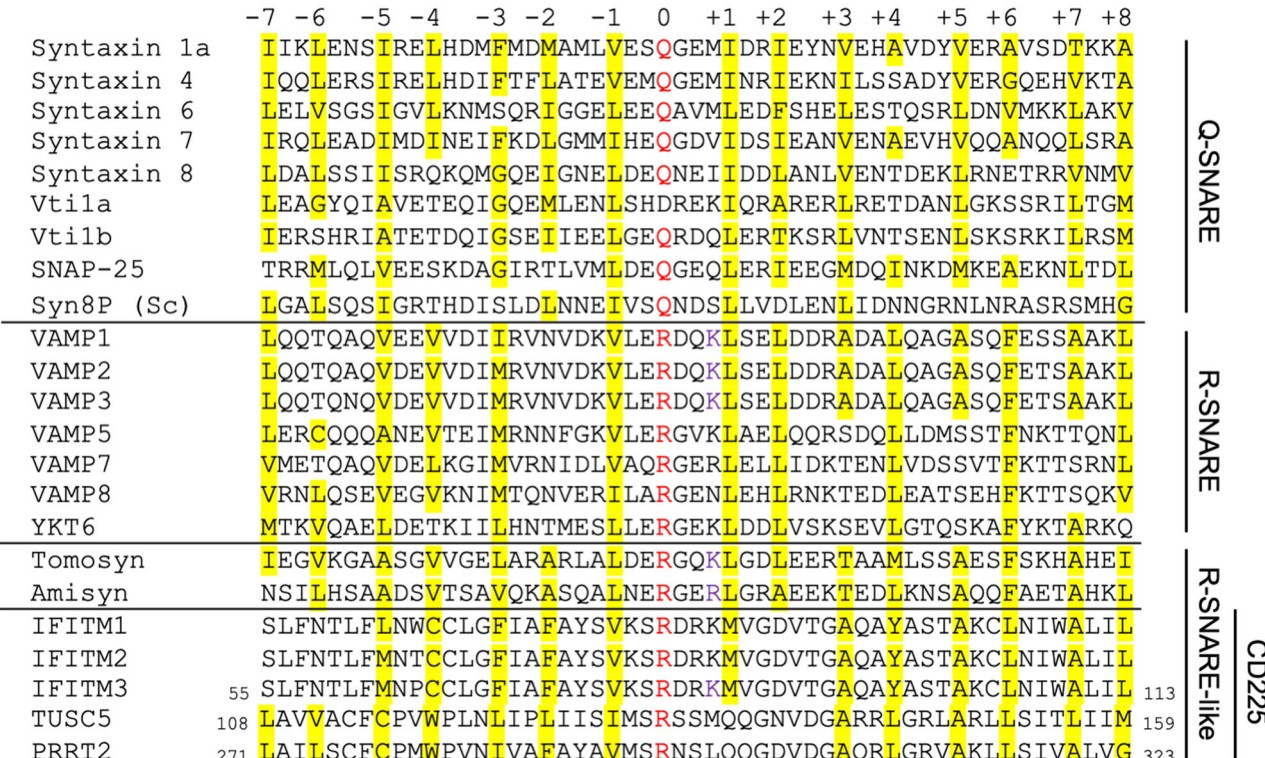

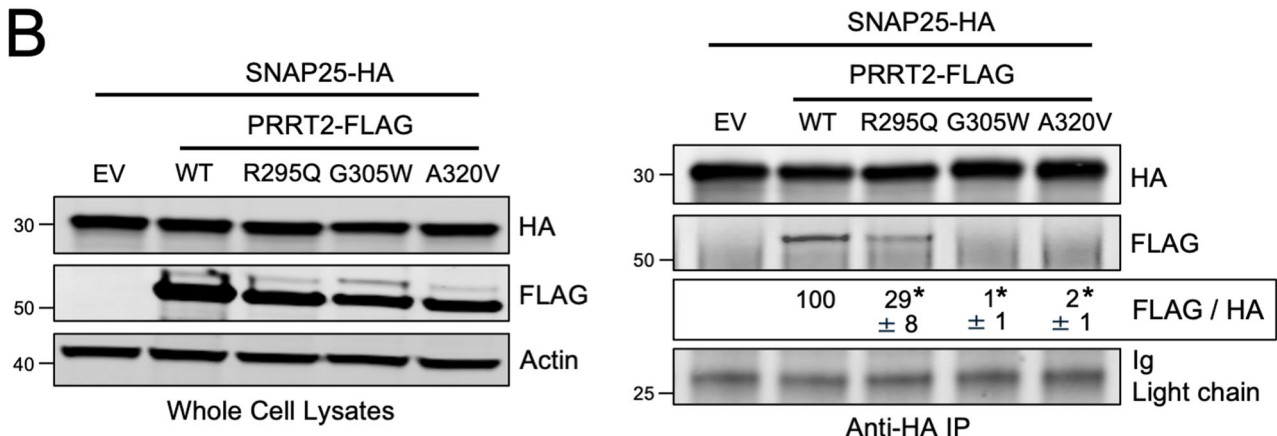

**Figure 1. CD225 proteins contain a motif resembling an R-SNARE motif.**

(A) A selection of SNARE proteins from yeast and mammals were aligned with proteins with SNARE-like motifs, and a 53-residue stretch of sequence is shown. Proteins are grouped by Q-SNARE, R-SNARE, or R-SNARE-like, and CD225 proteins are labeled as a subset of the R-SNARE-like proteins. Residues in red text are the central polar arginine or glutamine residues characteristic of canonical SNARE proteins and are labeled as the central "0" layer. Surrounding heptad repeats of hydrophobic residues are labeled from "−7" to "+8" in relation to the "0" layer, and sites highlighted in yellow correspond to hydrophobic residues. Underlined residues are those found in PRRT2, which when mutated, are associated with neurological dysfunction. (B) Left: HEK293T cells were co-transfected with SNAP25-HA and either PRRT2-FLAG (WT, R295Q, G305W, or A320V) or Empty Vector. SDS-PAGE and immunoblotting were performed with anti-HA and anti-FLAG in whole cell lysates. Anti-actin was used as loading control. Right: From co-transfected cells, SNAP25-HA was immunoprecipitated with anti-HA followed by SDS-PAGE and immunoblotting with anti-HA and anti-FLAG. Immunoglobulin light chain was used as loading control. The FLAG/HA ratio was calculated for the indicated lanes and shown as mean and standard error (normalized relative to WT, which was set to 100%). Differences that were statistically significant from WT as determined by one-way ANOVA are indicated by (*). Exact p values are as follows (from left to right): $p < 0.0001$, $p < 0.0001$, $p < 0.0001$. Immunoblots were performed independently three times (biological replicates), and a representative example is shown. Sc Saccharomyces cerevisiae. Ig immunoglobulin, IP immunoprecipitation, EV empty vector, WT wild-type. Source data are available online for this figure.

transmembrane protein 3 (IFITM3) (Coomer et al, 2021). TUSC5 and IFITM3 regulate vesicular trafficking during glucose transport and virus infection, respectively, through incompletely understood mechanisms. IFITM3 provides constitutive and interferon-induced protection of cells from endocytic viruses such as Influenza A virus (IAV) by homing to virus-containing endosomes and inhibiting virus-cell membrane fusion, effectively preventing the delivery of the viral core into host cell cytosol (where virus replication begins) (Desai et al, 2014; Klein et al, 2023; Li et al, 2013; Suddala et al, 2019). Moreover, *Ifitm3* knockout mice and disease-associated single-nucleotide polymorphisms in human *IFITM3* revealed that IFITM3 prevents severe viral disease sequelae in vivo (Bailey et al, 2012; Everitt et al, 2012).

Here, we report the identification of an R-SNARE-like motif shared among many CD225 family members that enables SNARE binding and modulation of SNARE assemblies. Mutation of residues within the R-SNARE-like motif of PRRT2, TUSC5, and IFITM3 resulted in reduced Q-SNARE binding and loss of their respective functions, as demonstrated experimentally or by associations with human disease. In-depth analysis revealed that IFITM3 interacts with endosomal SNARE proteins and, as a result, inhibits the assembly of the SNARE complex mediating homotypic late endosome fusion. Our findings suggest that cells expressing IFITM3 prioritize the heterotypic fusion pathway whereby late endosomes fuse with lysosomes, resulting in endosomal cargo degradation. Our findings demonstrate that CD225 proteins regulate diverse membrane trafficking processes in different tissues by regulating cellular fusion machinery via a conserved R-SNARE-like motif.

## Results

To investigate whether the known SNARE-binding activity exhibited by human PRRT2 may be attributed to the presence of a previously unrecognized SNARE-like motif, we performed multiple protein sequence alignment between PRRT2 and a panel of Q- and R-SNARE proteins. We found that PRRT2 encodes an arginine that aligns with the central arginines present within canonical R-SNARE motifs, which are ~53 amino acids in length (Fig. 1A). Furthermore, like canonical R-SNARE proteins, PRRT2 contains regularly spaced, generally hydrophobic residues extending on both sides of the central arginine. Therefore, PRRT2 contains a putative R-SNARE motif, resembling that which is present in VAMP proteins. Importantly, we also found this R-SNARE-like motif in additional members of the human CD225 protein family, including IFITM1, IFITM2, IFITM3, IFITM5, IFITM10, and TUSC5, although the identity and number of hydrophobic layers surrounding the central arginine varied between members (Fig. 1). The central arginine was lacking in a subset of CD225 family members (Fig. EV1A), suggesting either that the R-SNARE-like motif does not trace its origins to the CD225 ancestor or that the motif was lost to decay in certain family members.

Next, since PRRT2 has been described to regulate the SNARE complex controlling neurotransmission (Coleman et al, 2018), we measured whether the R-SNARE-like motif present within PRRT2 confers it with the ability to bind neuronal SNARE proteins. In agreement with previous reports, ectopic PRRT2 co-immunoprecipitated with ectopic Q-SNARE SNAP-25 (Fig. 1B).

Attesting to a critical role played by the R-SNARE-like motif of PRRT2 in mediating this interaction, mutation of the central arginine (R295Q) resulted in decreased co-immunoprecipitation between PRRT2 and SNAP-25 (Fig. 1B). Furthermore, mutation of one of the hydrophobic layers of the R-SNARE-like motif of PRRT2 (A320V) also abrogated the interaction with SNAP-25, and a G305W mutation that is adjacent to a hydrophobic layer and which inhibits PRRT2 homodimerization (Rahman et al, 2020) also abrogated this interaction (Fig. 1B). Meanwhile, these mutations in PRRT2 did not affect the interaction between PRRT2 and another Q-SNARE in the neuronal SNARE complex, syntaxin 1A (Fig. EV1B). Importantly, all three mutations (R295Q, G305W, and A320V) in PRRT2 occur naturally as polymorphisms in humans and are linked to neurological dysfunction (Becker et al, 2013; Ma et al, 2018; Tan et al, 2018). Therefore, our results describing an R-SNARE-like motif in human PRRT2 provide a mechanistic basis for the previously described molecular interaction between PRRT2 and neuronal SNARE proteins, as well as an explanation for how R295Q, G305W, and A320V cause loss of PRRT2 function in vivo and neurological dysfunction.

TUSC5 is a CD225 family member expressed in adipose tissue and implicated in the translocation of glucose transporter type 4 (GLUT4) to the cell surface, a process essential for insulin-stimulated glucose transport (St-Denis and Cushman, 1998; Stöckli et al, 2011). It was previously reported that TUSC5 co-immunoprecipitated with GLUT4 and SNARE proteins implicated in the vesicular trafficking of GLUT4 (Beaton et al, 2015; Fazakerley et al, 2015). However, the basis for this interaction was unknown. Like the CD225 family member PRRT2, TUSC5 also encodes a putative R-SNARE-like motif (Fig. 1A). Accordingly, we found that ectopic TUSC5 co-immunoprecipitated with ectopic syntaxin 4 (STX4), a Q-SNARE involved in GLUT4 translocation, while the G141W mutation in TUSC5 (analogous to G305W in PRRT2) disrupted this interaction (Fig. EV1C). Reverse co-immunoprecipitation confirmed this finding (Fig. EV1D). Collectively, these results highlight the presence of a R-SNARE-like motif in CD225 family members that enables coordination of SNARE proteins involved in diverse membrane fusion processes.

The IFITM proteins IFITM1, IFITM2, and IFITM3 are among the best characterized members of the CD225 protein family due to their established roles in cell-intrinsic defense against diverse pathogenic viruses (Majdoul and Compton, 2021; Shi et al, 2017). IFITM2 arose through gene duplication of IFITM3 in the ancestor of great apes, and both IFITM2 and IFITM3 proteins accumulate in membranes of the endolysosomal compartment (Compton et al, 2016). In addition to preventing virus escape from the endocytic compartment, it has been demonstrated that IFITM3 redirects incoming virions to lysosomes for degradation (Feeley et al, 2011; Spence et al, 2018; Suddala et al, 2019). However, the mechanism for this latter effect by IFITM3 was unknown, as was its relevance to the antiviral activity exhibited by IFITM3. Since IFITM3 is an endolysosomal protein, we assessed whether the R-SNARE-like motif in IFITM3 (Fig. 1A) conferred IFITM3 with the ability to bind and regulate endolysosomal SNARE proteins.

Fusion between compartments of the endolysosomal system are controlled by SNARE complexes consisting of the Q-SNAREs syntaxin 7 (STX7), syntaxin 8 (STX8), Vti1b, and one of two R-SNAREs—VAMP8 or VAMP7 (Fig. 2A). The STX7-STX8-Vti1b-VAMP8 complex enables late endosome-late endosome

   

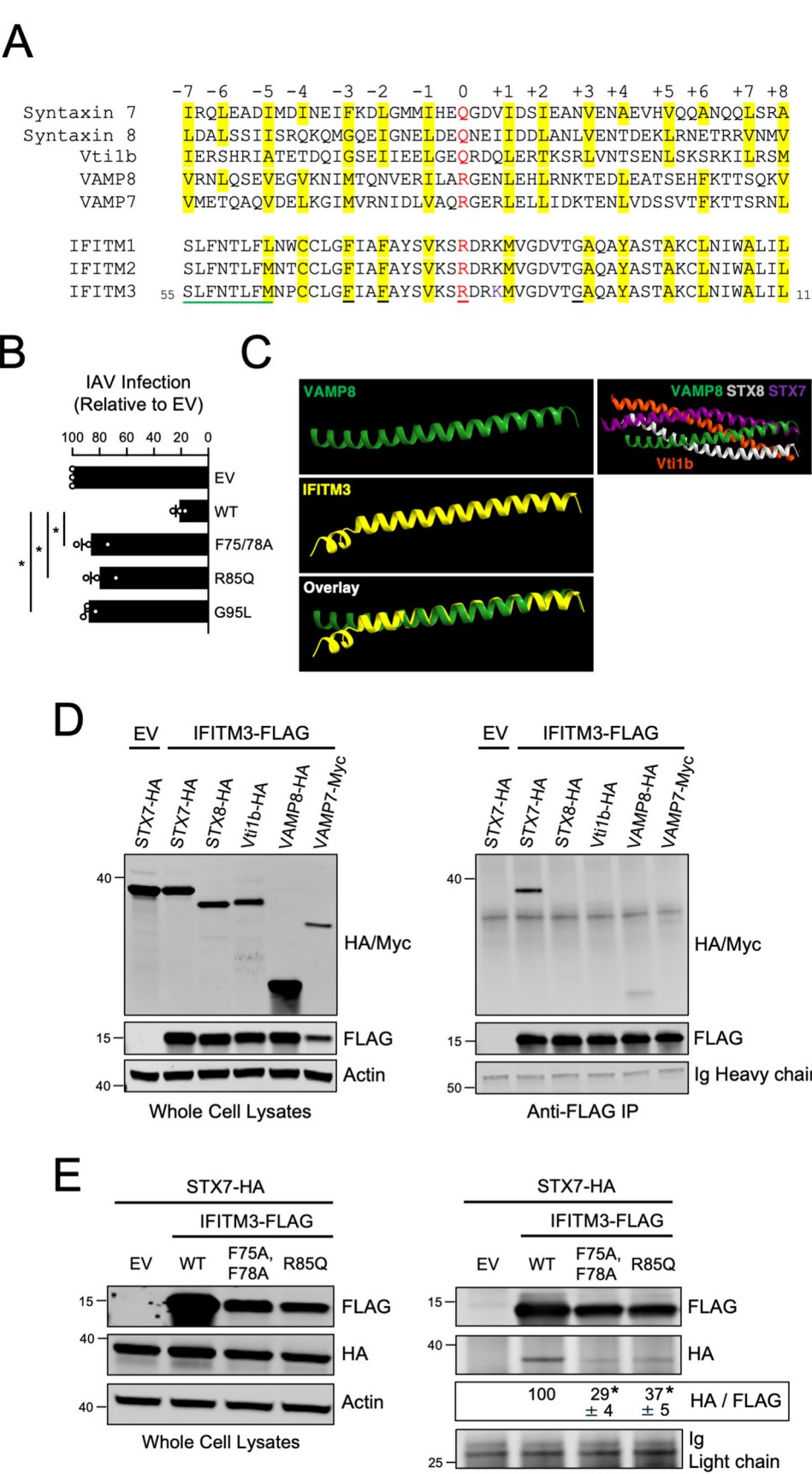

**Figure 2.  IFITM3 contains an R-SNARE-like motif and interacts with ectopic endosomal Q-SNARE STX7.**

(A) Human endosomal SNARE proteins were aligned with human IFITM proteins, and a 53-residue stretch of sequence is shown. Residues in red text are the central polar arginine or glutamine residues characteristic of canonical SNARE proteins and are labeled as the central "0" layer. Surrounding heptad repeats of hydrophobic residues are labeled from "−7" to "+8" in relation to the "0" layer, and sites highlighted in yellow correspond to hydrophobic residues. Black underlined residues in IFITM3 were mutated for functional experiments. Green underlined residues in IFITM3 correspond to the previously described amphipathic alpha helix. (B) HEK293T cells were transfected with IFITM3-FLAG (WT, F75/78A, R85Q, or G95L) and, 48 h later, challenged with IAV (A/PR/8/34 (PR8), H1N1) at a multiplicity of infection of 0.10. 18 h post-infection, cells were fixed, permeabilized, immunostained with anti-FLAG and anti-NP, and analyzed by flow cytometry. The percentage of NP+ cells was measured as a fraction of FLAG+ cells and shown as mean and standard error (normalized relative to Empty Vector, which was set to 100%). Infections were performed independently three times (biological replicates). Differences that were statistically significant from WT as determined by one-way ANOVA are indicated by (*). Exact *p* values are as follows (from left to right): $p < 0.0001$, $p < 0.0001$, $p < 0.0001$. (C) Structural prediction of the R-SNARE-like motif of IFITM3 was performed with Alphafold and FATCAT. Residues 55–113 of IFITM3 were modeled against a template of the R-SNARE motif of VAMP8 which was previously crystallized as part of the STX7-STX8-Vti1b-VAMP8 trans-SNARE complex (PDB: 1GL2). (D) Left: HEK293T cells were co-transfected with IFITM3-FLAG or Empty Vector and a construct encoding a single SNARE protein: STX7-HA, STX8-HA, Vti1b-HA, VAMP8-HA, or VAMP7-myc. SDS-PAGE and immunoblotting were performed with anti-HA, anti-myc, and anti-FLAG in whole cell lysates. Anti-actin was used as loading control. Right: From co-transfected cells, IFITM3-FLAG was immunoprecipitated with anti-FLAG followed by SDS-PAGE and immunoblotting with anti-HA, anti-myc, and anti-FLAG. Heavy chain immunoglobulin chain was used as loading control. Co-transfection of Empty Vector and STX7-HA was used as a negative control for immunoprecipitation. (E) Left: HEK293T cells were co-transfected with STX7-HA and either IFITM3-FLAG (WT, F75/78A, or R85Q) or Empty Vector. SDS-PAGE and immunoblotting were performed with anti-HA and anti-FLAG in whole cell lysates. Anti-actin was used as loading control. Right: From co-transfected cells, IFITM3-FLAG was immunoprecipitated with anti-FLAG followed by SDS-PAGE and immunoblotting with anti-FLAG and anti-HA. Light chain immunoglobulin was used as loading control. Co-transfection of Empty Vector and STX7-HA was used as a negative control for immunoprecipitation. The HA/FLAG ratio was calculated for the indicated lanes and shown as mean and standard error (normalized relative to WT, which was set to 100%). Differences that were statistically significant from WT as determined by one-way ANOVA are indicated by (*). Exact *p* values are as follows (from left to right): $p < 0.0001$, $p < 0.0001$. Numbers and tick marks left of blots indicate position and size (in kilodaltons) of protein standard in ladder. Immunoblots were performed independently three times, and a representative example is shown. IAV Influenza A virus, Ig immunoglobulin, IP immunoprecipitation, EV Empty Vector, WT wild-type. Source data are available online for this figure.

(homotypic) fusion while the STX7-STX8-Vti1b-VAMP7 complex is responsible for late endosome-lysosome (heterotypic) fusion (Advani et al, 1999; Mullock et al, 2000; Prekeris et al, 1999; Pryor et al, 2004). Endocytic viruses like IAV exploit the endolysosomal network and the SNARE machinery found therein in order to avoid lysosomal degradation and gain access to host cell cytosol, where virus replication is initiated. VAMP8 and its role in homotypic late endosome fusion was previously reported to promote IAV entry into cells. In contrast, VAMP7 inhibited IAV entry, presumably by favoring the degradation of virus-laden late endosomes with lysosomes (Pirooz et al, 2014a; Pirooz et al, 2014b). Therefore, we tested whether IFITM3 and the R-SNARE-like motif found in its CD225 domain could influence the assembly of SNARE complexes regulating the endolysosomal network.

The R-SNARE-like motif is conserved among the interferon-inducible members of the human IFITM family that exhibit antiviral activity—IFITM1, IFITM2, and IFITM3 (Fig. 2A). In contrast to a canonical R-SNARE protein like VAMP8, the R-SNARE-like motif of IFITM1-3 lack two layers of hydrophobicity at the amino-terminal end of the motif. Instead, IFITM1-3 encode an amphipathic helix that was shown to mediate cholesterol binding activity, membrane binding, and which is essential for antiviral activity against IAV (Chesarino et al, 2017; Guo et al, 2021; Rahman et al, 2020; Rahman et al, 2022) (Fig. 2A). Interestingly, the intact hydrophobic layers forming part of the R-SNARE-like motif downstream of the amphipathic helix include residues that were previously shown to be important for antiviral activity: a cysteine residue previously shown to be palmitoylated (Chesarino et al, 2014; Hach et al, 2013; Yount et al, 2010) and two phenylalanine residues shown to be critical for antiviral function (John et al, 2013; Suddala et al, 2019) (Fig. 2A). Therefore, to provide further evidence that the R-SNARE-like motif of IFITM3 contributes functionally to its antiviral activity, we assessed how mutation of the central arginine (R85Q) or the two phenylalanines (F75A/F78A) impacted antiviral activity against IAV. We also tested the effect of mutating glycine 95 (G95L), since it is adjacent to a hydrophobic layer in the R-SNARE-like motif of IFITM3, is homologous to glycine 305 in PRRT2 and glycine 141 in TUSC5

(Figs. 1B and EV1C,D), and was previously shown by us to be critical for antiviral activity (Rahman et al, 2020). While IFITM3 WT reduced IAV infection by ~80% relative to Empty Vector, all mutants tested exhibited a near-total loss of antiviral activity (Fig. 2B). These results indicate that the antiviral activity of IFITM3 against IAV depends upon a previously uncharacterized R-SNARE-like motif. Furthermore, our findings suggest that interactions with SNARE machinery may comprise an essential but unrecognized component of IFITM3 function in cells and may explain certain cellular phenotypes associated with IFITM3 overexpression (Zhong et al, 2022).

Next, we addressed whether the R-SNARE-like motif of IFITM3 adopts a protein fold resembling that of a genuine R-SNARE. Since the atomic structure of IFITM3 is incompletely understood (Ling et al, 2016) and the influence of interaction partners on its structure has not been explored, we used the TM-align platform (Zhang and Skolnick, 2005) to perform protein structure alignment between the R-SNARE-like motif of IFITM3 and the R-SNARE motif of VAMP8. The results supported that IFITM3 and VAMP8 adopt a highly similar alpha helical fold in the region spanning their respective R-SNARE motifs (TM-score = 0.76) (Fig. EV2A). To further investigate the SNARE-like structural qualities of IFITM3, we used Alphafold (Mirdita et al, 2022; Varadi et al, 2024) and the flexible protein structure alignment algorithm FATCAT (Li et al, 2020) to model the R-SNARE-like motif of IFITM3 against a template consisting of the R-SNARE motif of VAMP8, which was previously crystallized in the context of the STX8-STX7-Vti1b-VAMP8 complex (Antonin et al, 2002). We found that the Alphafold model for the R-SNARE-like motif of IFITM3 exhibited a high degree of structural homology with the R-SNARE motif of VAMP8 (Fig. 2C). We also swapped VAMP8 for IFITM3 in the context of the STX7-STX8-Vti1b-VAMP8 complex to provide spatial orientation of the F75, F78, and R85 residues of IFITM3. We found that the side chains of F75, F78, and R85 are predicted to radiate towards the center of the coiled-coil structure formed by Q-SNAREs STX7, STX8, and Vti1b (Fig. EV2B). Moreover, the central arginine (R85) of IFITM3 is predicted to align with the central glutamine (Q) residues of STX7, STX8, and Vti1b

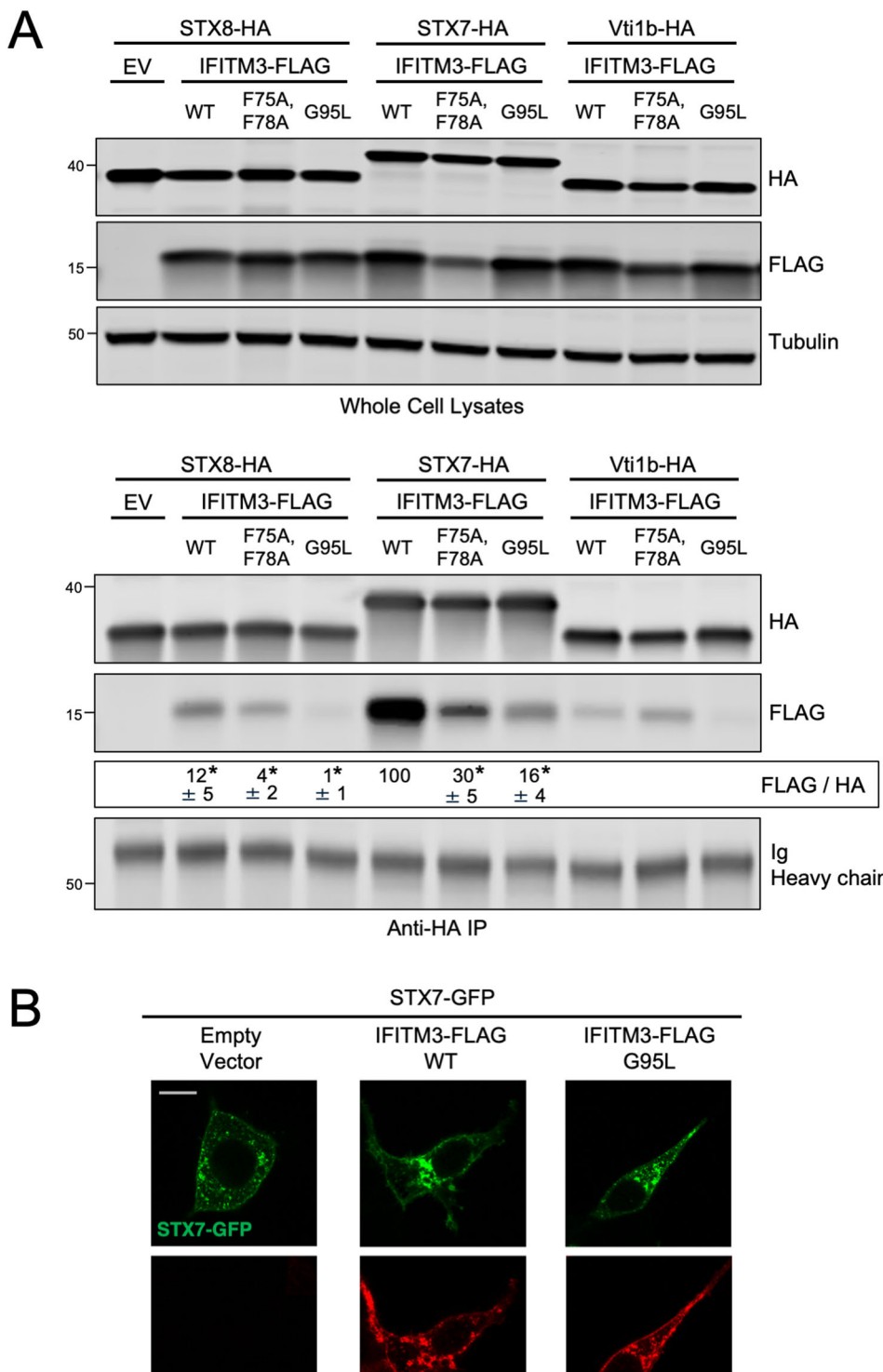

A

**Whole Cell Lysates**

**Anti-HA IP**

| | | | | | | | | | | | FLAG / HA |
|---|---|---|---|---|---|---|---|---|---|---|---|
| | 12* ± 5 | 4* ± 2 | 1* ± 1 | 100 | 30* ± 5 | 16* ± 4 | | | | | |

B

STX7-GFP

STX7 / FLAG
PCC = 0.580 ± 0.066

STX7 / FLAG
PCC = 0.155 ± 0.065

◄ **Figure 3. IFITM3 interacts with endosomal Q-SNAREs in a G95-dependent manner.**

(A) Top: HEK293T cells were co-transfected with STX8-HA, STX7-HA, or Vti1b-HA and either Empty Vector or IFITM3-FLAG (WT, F75/78A, or G95L). SDS-PAGE and immunoblotting were performed with anti-HA and anti-FLAG in whole cell lysates. Anti-tubulin was used as loading control. Bottom: from co-transfected cells, STX8-HA, STX7-HA, or Vti1b-HA were immunoprecipitated with anti-HA followed by SDS-PAGE and immunoblotting with anti-HA and anti-FLAG. Heavy chain immunoglobulin was used as loading control. Co-transfection of Empty Vector and STX8-HA was used as a negative control for immunoprecipitation. The FLAG/HA ratio was calculated for the indicated lanes and shown as mean and standard error (normalized relative to WT with STX7-HA, which was set to 100%). Differences that were statistically significant from WT with STX7-HA as determined by one-way ANOVA are indicated by (*). Exact $p$ values are as follows (from left to right): $p < 0.0001$, $p < 0.0001$, $p < 0.0001$, $p < 0.0001$, $p < 0.0001$. Numbers and tick marks left of blots indicate position and size (in kilodaltons) of protein standard in ladder. Immunoblots were performed independently three times, and a representative example is shown. (B) HEK293T cells stably expressing Empty Vector, IFITM3 WT-FLAG, or IFITM3 G95L-FLAG were transfected with STX7-GFP, fixed, immunostained with anti-FLAG, and analyzed by confocal immunofluorescence microscopy. Colocalization was measured between FLAG and STX7-GFP by calculating the Pearson's correlation coefficient using Fiji software. Coefficients were calculated from medial Z-slices from three fields of view containing 5–15 cells per condition and presented as means and standard error. Scale bar = 15 microns. Ig immunoglobulin, IP immunoprecipitation, EV Empty Vector, WT wild-type, PCC Pearson's correlation coefficient. Source data are available online for this figure.

(Fig. EV2C). Therefore, IFITM3 encodes an R-SNARE-like motif that is predicted to behave like an endosomal R-SNARE protein.

To determine whether the R-SNARE-like motif of IFITM3 confers it with the ability to bind endolysosomal SNARE proteins in cells, we overexpressed IFITM3 in combination with individual endolysosomal SNARE proteins in HEK293T cells and performed co-immunoprecipitation. We found that IFITM3 immunoprecipitated from transfected cells pulled down relatively high quantities of ectopic STX7, while other endolysosomal SNARE proteins were pulled down to a lesser, or negligible, extent (Fig. 2D). This selective co-immunoprecipitation between IFITM3 and STX7 was not due to elevated STX7 levels in transfected cells (Fig. 2D). Importantly, mutations in IFITM3 that caused loss of antiviral function (F75/78A, R85Q, and G95L) (Fig. 2B) diminished binding with STX7 relative to WT (Fig. 2E). IFITM3 F75/78A and R85Q exhibited somewhat decreased steady-state levels relative to WT, possibly attesting to the importance of the R-SNARE-like motif in maintaining proper folding and stability of the IFITM3 polypeptide (Fig. 2E). The association between IFITM3 and STX7 was confirmed by reverse co-immunoprecipitation, whereby immunoprecipitated ectopic STX7 pulled down ectopic IFITM3 (Fig. 3A). Ectopic STX8 also co-immunoprecipitated with IFITM3 but to a lesser extent than STX7 (Fig. 3A). To further support an association between IFITM3 and STX7, we performed confocal immunofluorescence in intact cells transfected with ectopic IFITM3 and STX7-GFP. We observed colocalization between IFITM3 WT and STX7-GFP, while IFITM3 G95L exhibited decreased colocalization (Fig. 3B). These results demonstrate that an R-SNARE-like motif in IFITM3 enables it to bind to endosomal SNARE proteins, particularly Q-SNAREs STX7 and STX8. Furthermore, there exists a functional requirement for the R-SNARE-like motif of IFITM3 during restriction of IAV infection.

While our results are suggestive of selective binding between IFITM3 and Q-SNAREs STX7 and STX8, we also detected pull down of VAMP8-HA when IFITM3 was immunoprecipitated (Fig. 2D). To confirm this interaction, we performed the reverse co-immunoprecipitation and found that VAMP8-HA pulled down IFITM3 WT, while pulldown was reduced by the G95L mutation (Fig. EV3A). We also immunoprecipitated VAMP7-Myc and assessed pull down of ectopic IFITM3, because a previously published proteomic dataset identified VAMP7 as a possible binding partner of IFITM3 (Li et al, 2024). As was observed for VAMP8-HA, VAMP7-Myc pulled down IFITM3 WT, while IFITM3 G95L was pulled down to a lesser extent (Fig. EV3B). Together, these results suggest that ectopic IFITM3 can interact

with endosomal Q-SNAREs as well as the R-SNAREs VAMP8 and VAMP7.

We then assessed whether endogenous IFITM3 exhibits SNARE binding activity. Following transfection of ectopic STX7 into HeLa WT or *IFITM3* KO cells, we found that ectopic STX7 pulled down endogenous IFITM3 (Fig. 4A). As a control, pull down of endogenous IFITM3 by ectopic STX7 was lost in *IFITM3* KO cells (Fig. 3A). *IFITM3* KO was specific, as IFITM2 protein levels were unaffected, and did not influence the overall levels of endolysosomal SNARE proteins (Fig. EV4A). Moreover, the presence or absence of IFITM3 did not affect the co-immunoprecipitation between STX7 and other Q-SNAREs STX8 and Vti1b (Fig. EV4B). We were unable to detect or immunoprecipitate endogenous STX7 in HeLa cells. However, we succeeded in immunoprecipitating endogenous STX8 and found that it pulled down endogenous IFITM3 (Fig. 4B). In addition, endogenous IFITM3 was also pulled down by endogenous Vti1b, VAMP8, and VAMP7 (Fig. 4C–E). These results demonstrate that endogenous IFITM3 binds to endolysosomal Q- and R-SNAREs in a direct or indirect manner. To determine whether IFITM3 can bind SNARE proteins directly, we produced and purified recombinant IFITM3 (Fig. EV4C,D) and incubated it with recombinant STX7 in vitro (Fig. EV4E,F). Immunoprecipitation of recombinant STX7 from the reaction resulted in pull down of recombinant IFITM3 WT, but this interaction was lost for IFITM3 F75/78A or IFITM3 G95L (Fig. 4F). With size exclusion chromatography, we ruled out that the selective pull down of IFITM3 WT by recombinant STX7 was due to a tendency for IFITM3 WT to form aggregates in vitro (Fig. EV5A). Overall, we demonstrate for the first time that IFITM3 is capable of direct interaction with an endosomal Q-SNARE via an R-SNARE-like motif that is critical for antiviral activity.

Since STX7 is an essential part of the SNARE complexes controlling homotypic late endosome fusion and heterotypic late endosome fusion, we determined whether IFITM3 regulates the assembly of the STX7-STX8-Vti1b-VAMP8 complex, the STX7-STX8-Vti1b-VAMP7 complex, or both. STX7, VAMP8, and IFITM3 were ectopically co-expressed in HEK293T and STX7 was immunoprecipitated to identify its interaction partners. As shown above, STX7 pulled down IFITM3 WT, while the G95L mutation in IFITM3 abrogated this interaction (Fig. 5A). Importantly, STX7 pulled down a reduced amount of VAMP8 in the presence of IFITM3 relative to Empty Vector. In contrast, IFITM3 G95L did not affect the STX7-VAMP8 interaction (Fig. 5A). These findings were confirmed by measuring the colocalization between STX7-GFP and VAMP8-mCherry in cells expressing IFITM3 WT

    

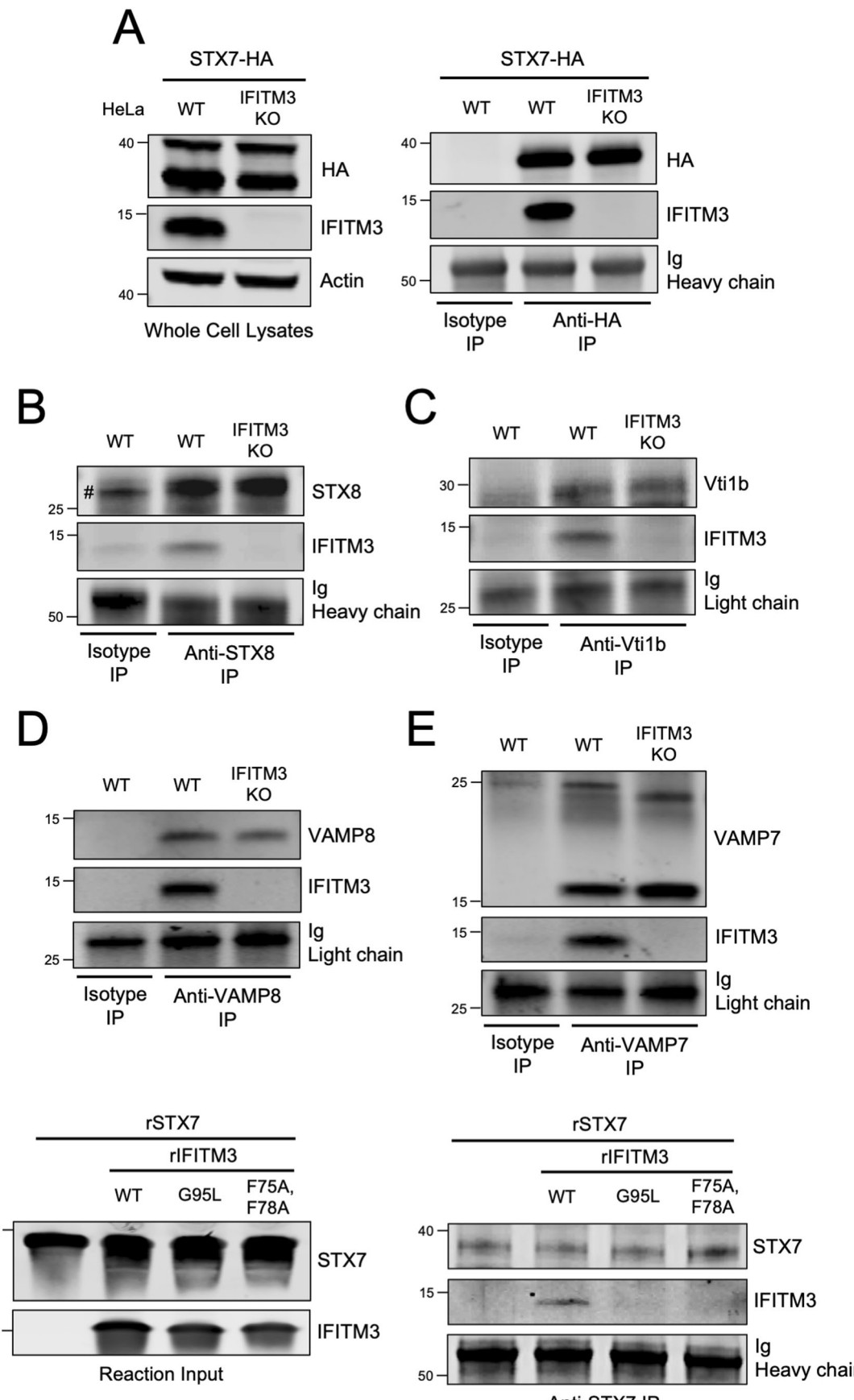

**Figure 4. Endogenous IFITM3 interacts with endosomal Q-SNAREs and R-SNAREs via an R-SNARE-like motif.**

(A) Left: HeLa cells (WT or *IFITM3* KO) were transfected with STX7-HA. SDS-PAGE and immunoblotting were performed with anti-HA and anti-IFITM3 in whole cell lysates. Anti-actin was used as loading control. Right: STX7-HA was immunoprecipitated with anti-HA antibody followed by SDS-PAGE and immunoblotting with anti-HA and anti-IFITM3. Heavy chain immunoglobulin was used as loading control. An isotype matched antibody was used as a control for immunoprecipitation. (B) Endogenous STX8 was immunoprecipitated from HeLa (WT or *IFITM3* KO) with anti-STX8 followed by SDS-PAGE and immunoblotting with anti-STX8 and anti-IFITM3. Heavy chain immunoglobulin was used as loading control. An isotype matched antibody was used as a control for immunoprecipitation. The (#) symbol denotes the presence of light chain immunoglobulin. (C) Endogenous Vti1b was immunoprecipitated from HeLa (WT or *IFITM3* KO) with anti-Vti1b followed by SDS-PAGE and immunoblotting with anti-Vti1b and anti-IFITM3. Light chain immunoglobulin was used as loading control. (D) Endogenous VAMP8 was immunoprecipitated from HeLa (WT or *IFITM3* KO) with anti-VAMP8 followed by SDS-PAGE and immunoblotting with anti-VAMP8 and anti-IFITM3. Light chain immunoglobulin was used as loading control. (E) Endogenous VAMP7 was immunoprecipitated from HeLa (WT or *IFITM3* KO) with anti-VAMP7 followed by SDS-PAGE and immunoblotting with anti-VAMP7 and anti-IFITM3. Two forms of VAMP7 were detected following immunoprecipitation: one form of ~25 kD and another of ~15 kD (Wojnacki et al, 2021). Light chain immunoglobulin was used as loading control. (F) Left: Recombinant STX7 and recombinant IFITM3 (WT, F75/78A, or G95L) were mixed together in vitro and reaction inputs were visualized by SDS-PAGE and immunoblotting with anti-STX7 and anti-IFITM3. Right: recombinant STX7 was immunoprecipitated with anti-STX7 followed by SDS-PAGE and immunoblotting with anti-STX7 and anti-IFITM3. Heavy chain immunoglobulin was used as loading control. Numbers and tick marks left of blots indicate position and size (in kilodaltons) of protein standard in ladder. Immunoblots were performed independently two times and a representative example is shown. Ig immunoglobulin, IP immunoprecipitation, KO knockout, WT wild-type, r recombinant. Source data are available online for this figure.

or IFITM3 G95L. We found that STX7-GFP colocalization with VAMP8-mCherry was reduced by IFITM3 WT in a G95-dependent manner (Fig. 5B). We performed additional experiments whereby STX7, VAMP7, and IFITM3 were ectopically co-expressed. Here, IFITM3 binding to STX7 did not inhibit the interaction between STX7 and VAMP7. Instead, the STX7-VAMP7 interaction was slightly elevated in the presence of IFITM3 WT, but not IFITM3 G95L (Fig. EV5B). These results indicate that IFITM3 binding to STX7 selectively impairs the interaction between STX7 and VAMP8.

Since it was previously shown that endocytic viruses including IAV may co-opt the STX7-STX8-Vti1b-VAMP8 SNARE complex to promote the virus entry process (Pirooz et al, 2014b), we performed co-immunoprecipitation experiments in cells that had been inoculated with IAV. Interestingly, we noticed that virus addition resulted in enhanced co-immunoprecipitation between endogenous STX8 and VAMP8 in HEK293T cells (Fig. 5C). However, in the presence of ectopic IFITM3 WT, this co-immunoprecipitation was diminished, while co-immunoprecipitation between endogenous STX8 and VAMP7 was enhanced (Fig. 5C). The IFITM3 G95L mutant, in contrast, co-immunoprecipitated with STX8 to a lesser extent than IFITM3 WT and only marginally impacted the STX8-VAMP8 and STX8-VAMP7 interactions (Fig. 5C). To address whether endogenous IFITM3 performs a similar function, we immunoprecipitated endogenous STX8 from HeLa. As shown above, endogenous IFITM3 pulls down with endogenous STX8. However, endogenous STX8 pulled down increased quantities of VAMP8 in *IFITM3* KO cells relative to WT cells (Fig. 5D). Consistently, STX7-GFP colocalized with VAMP8-mCherry to a greater extent in *IFITM3* KO cells compared to WT cells (Fig. EV5C). These results support a model whereby IFITM3 disfavors the STX7/8-VAMP8 interaction while favoring the STX7/8-VAMP7 interaction. To test whether IFITM3 reduces the assembly of the entire STX7-STX8-Vti1b-VAMP8 SNARE complex involved in homotypic late endosome fusion, we ectopically co-expressed STX7, STX8, Vti1b, and VAMP8 in HEK293T cells. SNARE protein assemblies have been previously characterized as SDS-resistant complexes in whole cell lysates under mildly denaturing conditions (Kubista et al, 2004). We identified an SDS-resistant complex greater than 140 kD in size by immunoblotting for VAMP8 (Fig. 6A) or STX8 (Appendix Fig. S1). The complex was sensitive to heat, as high molecular weight bands disappeared after boiling lysates (Fig. 6A; Appendix Fig. S1). Consistent with IFITM3 acting as a negative regulator of complex formation, complex abundance was

markedly reduced in cells expressing IFITM3 WT (Fig. 6A; Appendix Fig. S1). In contrast, IFITM3 G95L only marginally reduced complex formation (Fig. 6A). Therefore, by binding STX7/8 via an R-SNARE-like motif and reducing their interaction with VAMP8, IFITM3 impedes formation of the STX7-STX8-Vti1b-VAMP8 complex involved in homotypic late endosome fusion.

Our results showing that IFITM3 uses an R-SNARE-like motif to inhibit assembly of the homotypic late endosome fusion machinery while maintaining, and possibly amplifying, assembly of the heterotypic late endosome-lysosome fusion machinery may explain how IFITM3 redirects incoming virions towards lysosomes for degradation (Feeley et al, 2011; Spence et al, 2018; Suddala et al, 2019). To directly test whether IFITM3 promotes the fusion of late endosomes with lysosomes in living cells, we pulsed cells expressing IFITM3 WT or IFITM3 G95L with Dextran and monitored its trafficking to lysosomes labeled with Magic Red cathepsin substrate. In the presence of IFITM3 WT, Dextran colocalization with lysosomes was enhanced relative to cells expressing Empty Vector or IFITM3 G95L (Fig. 6B). When the same assay was performed in HeLa cells, Dextran colocalization with Magic Red was diminished in *IFITM3* KO cells compared to WT cells (Appendix Fig. S1B). As an alternative approach to measure late endosome-lysosome fusion, we co-transfected cells with Rab7-mCherry and LAMP1-GFP, which label late endosomes and lysosomes, respectively. In cells expressing IFITM3 WT, we observed increased colocalization between Rab7-mCherry and LAMP1-GFP relative to cells expressing Empty Vector or IFITM3 G95L (Appendix Fig. S2). In contrast, we did not observe extensive colocalization between Rab5-RFP (an early endosome marker) and LAMP1-GFP, and IFITM3 WT did not promote their colocalization (Appendix Fig. S3). Furthermore, the LAMP1-GFP compartment appeared enlarged and expanded in the presence of IFITM3 WT, which is consistent with enhanced late endosome-lysosome fusion (Appendix Figs. S2 and S3). While these findings demonstrate that IFITM3 promotes cargo delivery from late endosomes to lysosomes, it remained to be seen whether it promotes endosomal cargo degradation in lysosomes. It was previously reported that IFITM3 accelerates the turnover of the Epidermal Growth Factor Receptor (EGFR) in lysosomes (Spence et al, 2018). Importantly, the fate of EGFR in cells is also controlled by the STX7-STX8-Vti1b-VAMP7/8 SNARE machinery (Futter et al, 1996). To confirm this finding, we took advantage of an assay monitoring the lysosomal turnover of EGFR-GFP by flow cytometry (Inoue et al, 2015). We transfected EGFR-GFP into cells expressing Empty Vector, IFITM3 WT, or IFITM3 G95L and treated them with EGF to trigger

 

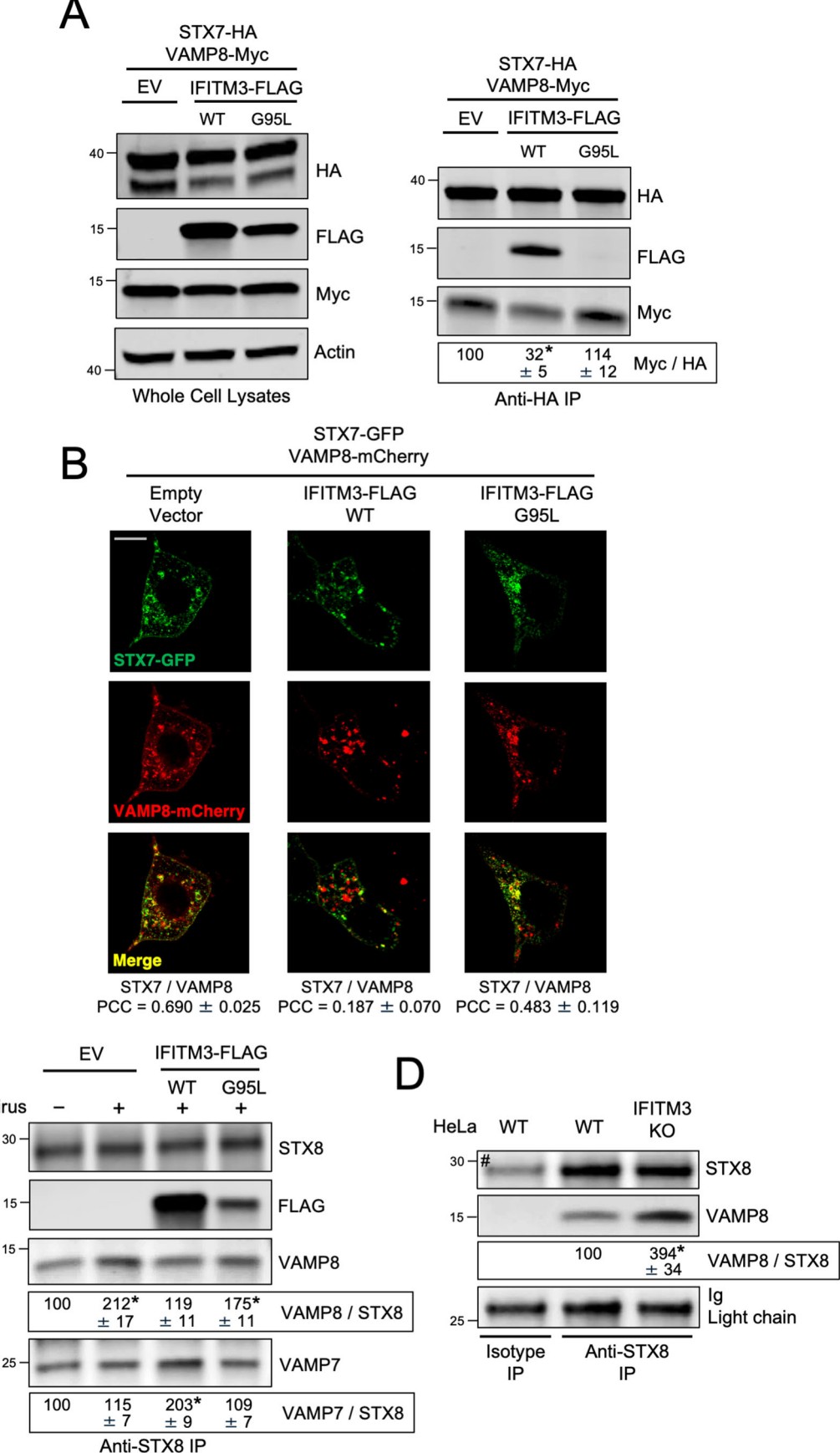

◄ **Figure 5. IFITM3 inhibits the interaction between STX7/8 and VAMP8.**

(A) HEK293T cells stably expressing Empty Vector, IFITM3 WT-FLAG, or IFITM3 G95L-FLAG were co-transfected with STX7-HA and VAMP8-Myc. Whole cell lysates were subjected to SDS-PAGE and immunoblotting with anti-HA, anti-FLAG, and anti-Myc. Actin was used as a loading control. Right: STX7-HA was immunoprecipitated with anti-HA followed by SDS-PAGE and immunoblotting with anti-HA, anti-FLAG, and anti-Myc. The Myc/HA ratio was calculated for the indicated lanes and shown as mean and standard error (normalized relative to Empty Vector, which was set to 100%). Differences that were statistically significant from Empty Vector as determined by one-way ANOVA are indicated by (*). Exact $p$ values are as follows (from left to right): $p = 0.0016$, $p = 0.4318$. (B) HEK293T cells stably expressing Empty Vector, IFITM3 WT-FLAG, or IFITM3 G95L-FLAG were transfected with STX7-GFP and VAMP8-mCherry, fixed, and analyzed by confocal immunofluorescence microscopy. Colocalization was measured between STX7-GFP and VAMP8-mCherry by calculating the Pearson's correlation coefficient using Fiji software. Coefficients were calculated from medial Z-slices from three fields of view containing 5–15 cells per condition and presented as means and standard error. Scale bar = 15 microns. (C) HEK293T cells stably expressing Empty Vector, IFITM3 WT-FLAG, or IFITM3 G95L-FLAG were inoculated with IAV (+) or medium (−) for 18 h and subjected to whole cell lysis. Endogenous STX8 was immunoprecipitated with anti-STX8 followed by SDS-PAGE and immunoblotting with anti-STX8, anti-FLAG, anti-VAMP8, and anti-VAMP7. The VAMP8/STX8 and VAMP7/STX8 ratios were calculated for the indicated lanes and shown as mean and standard error (normalized relative to No Virus Empty Vector, which was set to 100%). Differences that were statistically significant from No Virus Empty Vector as determined by one-way ANOVA are indicated by (*). Exact $p$ values are as follows (from left to right, top to bottom): $p = 0.0007$, $p = 0.6986$, $p = 0.0093$, $p = 0.4523$, $p < 0.0001$, $p = 0.7960$. (D) Endogenous STX8 was immunoprecipitated from HeLa cells (WT or *IFITM3* KO) with anti-STX8 followed by SDS-PAGE and immunoblotting with anti-STX8, anti-IFITM3, and anti-VAMP8. Light chain immunoglobulin was used as loading control. An isotype matched antibody was used as a control for immunoprecipitation. The VAMP8/STX8 ratio was calculated for the indicated lanes and shown as mean and standard error (normalized relative to WT, which was set to 100%). Differences that were statistically significant from WT as determined by student's T test are indicated by (*). Exact $p$ value: $p = 0.0009$. Numbers and tick marks left of blots indicate position and size (in kilodaltons) of protein standard in ladder. The (#) symbol denotes the presence of light chain immunoglobulin. Immunoblots were performed independently three times, and a representative example is shown. Ig immunoglobulin, IP immunoprecipitation, EV Empty Vector, WT wild-type, PCC Pearson's correlation coefficient. Source data are available online for this figure.

endocytosis of EGFR-GFP. Under these conditions, a portion of EGFR-GFP is slated for degradation in lysosomes. We found that partial EGFR-GFP loss was detected in HeLa WT cells between 1 and 2 h post-EGF addition. In contrast, we did not detect EGFR-GFP loss in HeLa *IFITM3* KO over the same time period (Appendix Fig. S4). Confirming a role for IFITM3 in accelerating the turnover of EGFR-GFP, we found that EGFR-GFP loss was less marked and less rapid in cells expressing IFITM3 G95L compared to cells expressing IFITM3 WT (Appendix Fig. S4). Therefore, IFITM3 facilitates the degradation of endosomal cargo by inhibiting assembly of the homotypic late endosome fusion machinery and promoting fusion between late endosomes and lysosomes.

To test the impact of IFITM3 on lysosomal targeting of viral cargo, we examined early steps of IAV entry into cells using confocal immunofluorescence microscopy. In HEK293T cells stably expressing IFITM3 WT or IFITM3 G95L, which differ with respect to STX7 binding potential (Fig. 3A), IAV was added to cells at a multiplicity of infection of 1 and incubated on ice for 40 min. Cells were fed warm medium containing Lysotracker, to visualize acidic lysosomes, and placed at 37 °C to enable synchronized internalization of virus particles. Four hours later, cells were washed, fixed, and immunostained with an antibody against IAV nucleoprotein (NP), which detects incoming, virion-associated NP as well as de novo synthesized NP. In cells expressing Empty Vector, nuclear NP accumulation was apparent at 4 h post-inoculation in roughly 20% of cells, which signals the initiation of virus replication following successful entry of the virus (i.e., escape from the endocytic compartment) (Fig. 7A, B). In cells expressing IFITM3 WT, nuclear NP accumulation was virtually absent, consistent with strong inhibition of cellular entry of IAV as reported previously (Feeley et al, 2011). In contrast, IFITM3 G95L did not reduce nuclear NP accumulation, in agreement with a loss of antiviral activity exhibited by this mutant (Rahman et al, 2020) (Fig. 7A,B). Interestingly, we observed a negative correlation between the extent of nuclear NP staining and the intensity of Lysotracker staining in cells. IFITM3 WT expression resulted in intensification of Lysotracker fluorescence, consistent with expansion of the lysosomal compartment as described previously (Feeley et al, 2011). However, expression of IFITM3 G95L did not (Fig. 7C,D). In cells expressing IFITM3 G95L, most NP protein localized to the nucleus at 4 h post-inoculation. However, some instances of extra-nuclear, bright NP

puncta associated with the Lysotracker-positive compartment could be observed, suggesting that a small proportion of incoming IAV virions were trafficked to lysosomes in IFITM3 G95L-expressing cells (Fig. 7D). However, the occurrence of bright NP puncta colocalizing with Lysotracker-positive vesicles was enhanced in cells expressing IFITM3 WT (Fig. 7D; Appendix Fig. S4). In IFITM3 WT cells, NP was found to colocalize with Lysotracker to a greater extent than in cells expressing IFITM3 G95L (Fig. 7D). These results support the notion that inhibition of homotypic late endosome fusion and promotion of heterotypic late endosome fusion by IFITM3 underlies its ability to promote trafficking of IAV towards lysosomes (Fig. 8). Altogether, this body of work demonstrates that endolysosomal SNARE modulation is a key aspect of the mechanism by which IFITM3 protects cells from invading viruses.

# Discussion

Our identification of an R-SNARE-like motif in human CD225 proteins suggests that these proteins act as SNARE mimics to regulate SNARE-mediated membrane fusion events. Even though there are CD225 members that lack the polar arginine or glutamine in the central layer characteristic of a canonical SNARE motif, they encode layered hydrophobicity which may still afford some degree of SNARE modulation (Fig. EV1A). For example, Vti1a is SNARE protein involved in neurotransmission despite encoding an aspartic acid in the central layer (Tang et al, 2022) (Fig. 1A). Moreover, other well-characterized SNARE proteins, like STX8, VAMP8, VAMP7, and Vti1b lack a full complement of hydrophobic layers in their respective SNARE motifs (Fig. 1A). Since the CD225 family is poorly characterized and contains many proteins with currently unknown functions (Coomer et al, 2021), our discovery of built-in SNARE binding potential within the CD225 domain may serve as a catalyst for uncovering the physiological roles they perform. As such, we provide important insight into the birth and functional expansion of the CD225 domain over the course of evolutionary time.

A limited number of SNARE-like proteins functioning as SNARE mimics have been described before. Two of the best characterized R-SNARE-like proteins in humans are Tomosyn and Amisyn, which adopt VAMP-like protein folds and regulate the same neuronal

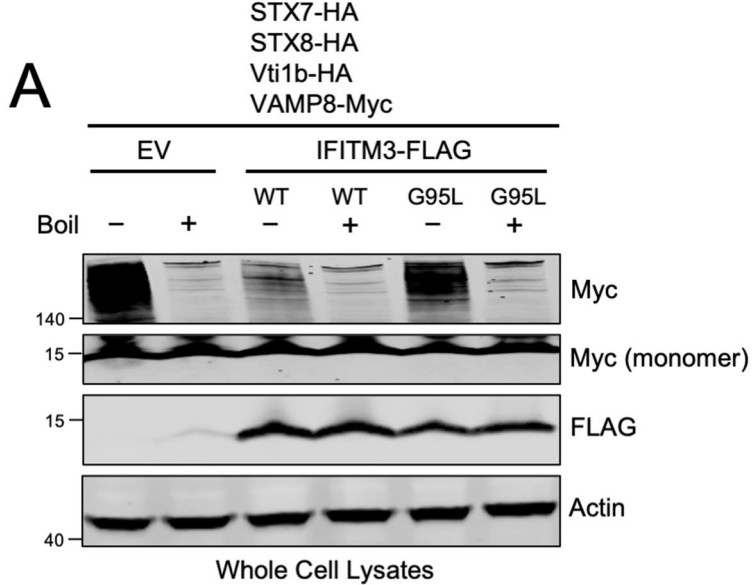

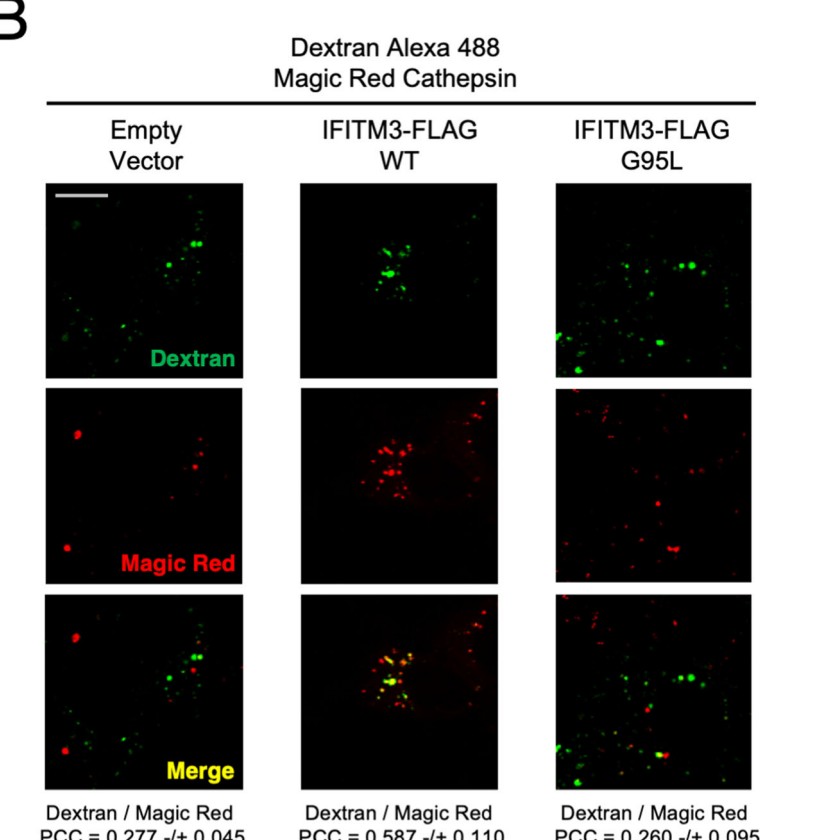

**Figure 6. IFITM3 inhibits assembly of homotypic late endosome fusion machinery and promotes the delivery of endosomal cargo to lysosomes.**

(**A**) HEK293T stably expressing Empty Vector, IFITM3 WT-FLAG, or IFITM3 G95L-FLAG were co-transfected with STX7-HA, STX8-HA, Vti1b-HA, and VAMP8-Myc. Following whole cell lysis, samples were either boiled at 100 °C (+) or not (−). SDS-PAGE and immunoblotting was performed with anti-Myc and anti-FLAG. Actin was used as loading control. Numbers and tick marks left of blots indicate position and size (in kilodaltons) of protein standard in ladder. Immunoblots were performed independently twice, and a representative example is shown. (**B**) HEK293T cells stably expressing Empty Vector, IFITM3 WT-FLAG, or IFITM3 G95L-FLAG were pulsed with Dextran Alexa Fluor 488 for 2 h followed by addition of Magic Red for 5 min. Living cells were then analyzed immediately by confocal immunofluorescence microscopy. Colocalization between Dextran and Magic Red was measured by calculating the Pearson's correlation coefficient using Fiji software. Coefficients were calculated from medial Z-slices from three fields of view containing 5–15 cells per condition and presented as means and standard error. Scale bar = 15 microns. PCC Pearson's correlation coefficient. Source data are available online for this figure.

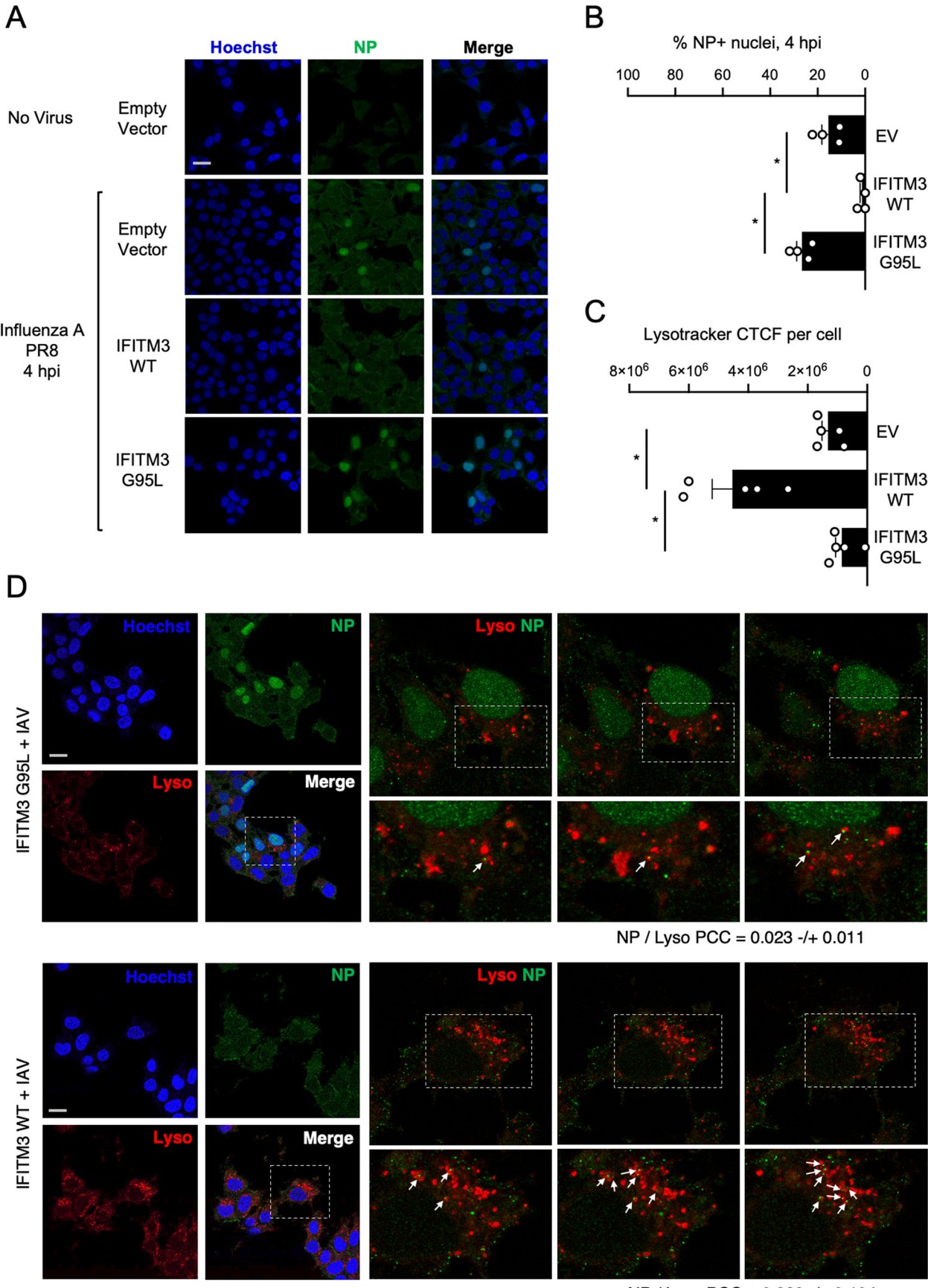

NP / Lyso PCC = 0.023 -/+ 0.011

NP / Lyso PCC = 0.389 -/+ 0.104

**Figure 7. IFITM3 traffics incoming IAV to lysosomes in a G95-dependent manner.**

(A) HEK293T cells stably expressing Empty Vector, IFITM3 WT-FLAG, or IFITM3 G95L-FLAG inoculated with IAV at a multiplicity of infection of 1 and incubated on ice for 40 min. Cells were washed, fresh medium containing Lysotracker Deep Red (50 nM) was added, and cells were placed at 37 °C. Four hours later, cells were fixed and immunostained with anti-NP antibody and Hoechst and analyzed by confocal immunofluorescence microscopy. Mock-inoculated (PBS) Empty Vector cells were used as a negative control. Scale bar = 15 microns. Medial Z-slices representative of each condition are shown. (B) The percentage of NP+ nuclei was quantified in Empty Vector, IFITM3 WT-FLAG, or IFITM3 G95L-FLAG cells. Symbols represent fields of view containing ~15–30 total cells and mean percentage plus standard error is shown. Differences that were statistically significant between the indicated conditions as determined by one-way ANOVA are indicated by (*). Exact p values are as follows (from left to right): $p < 0.0001$, $p = 0.0028$. (C) The corrected total cell fluorescence (CTCF) intensity of Lysotracker staining was quantified in Empty Vector, IFITM3 WT-FLAG, or IFITM3 G95L-FLAG cells. Symbols represent fields of view containing ~15–30 total cells and mean CTCF intensity plus standard error is shown. Differences that were statistically significant between the indicated conditions as determined by one-way ANOVA are indicated by (*). Exact p values are as follows (from left to right): $p = 0.0005$, $p = 0.0001$. (D) Examination of NP signal in relation to Lysotracker staining in HEK293T stably expressing IFITM3 WT-FLAG or IFITM3 G95L-FLAG. White boxes indicate fields of view selected for detailed analysis. Right: detailed view of three different Z slices from confocal stack. White arrows indicate bright NP puncta that colocalize with Lysotracker or which tightly appose a Lysotracker+ compartment. Colocalization between NP and Lysotracker was measured by calculating the Pearson's correlation coefficient using Fiji software. Coefficients were calculated from medial Z-slices from three fields of view containing 5–15 cells per condition and presented as means plus standard error. Scale bar = 15 microns. EV Empty Vector, WT wild-type, hpi hours post-inoculation, CTCF corrected total cell fluorescence, PCC Pearson's correlation coefficient. Source data are available online for this figure.

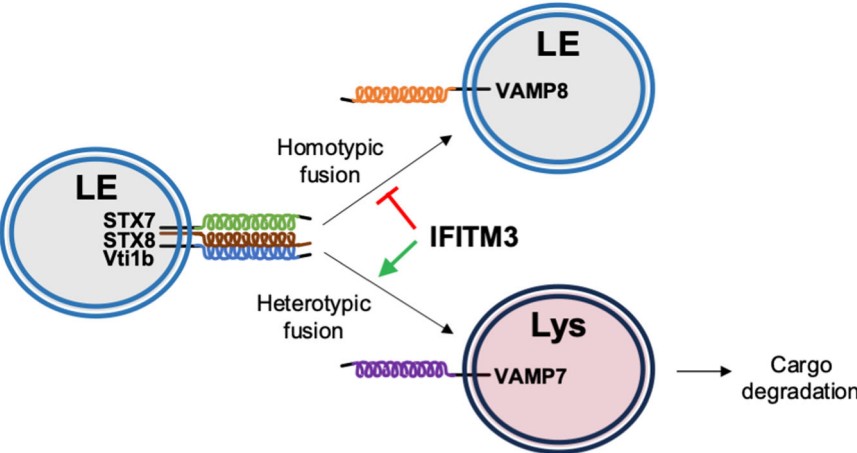

**Figure 8. A model of inhibition of homotypic late endosome fusion by IFITM3.**

IFITM3 selectively regulates the trans-SNARE assembly driving late endosome-late endosome fusion (STX7-STX8-Vti1b-VAMP8), while enabling late endosome-lysosome fusion. As a result, endocytic cargos (including viruses) are more efficiently trafficked to lysosomes in IFITM3-expressing cells. LE late endosome, Lys lysosome.

SNARE complex that PRRT2 does (Hatsuzawa et al, 2003; Masuda et al, 1998; Scales et al, 2002). Furthermore, microbial proteins containing SNARE-like sequence and functionality have been discovered in prokaryotes and viruses (Delevoye et al, 2008; King et al, 2015; Paumet et al, 2009; Shi et al, 2016; Wesolowski and Paumet, 2010). Interestingly, phylogenetic evidence suggests that the CD225 domain was introduced into eukaryotes following multiple horizontal transmission events from prokaryotes (Sällman Almén et al, 2012). Therefore, it is possible that the SNARE-like motif of CD225 traces its beginnings to a similar, ancient motif in microbes.

IFITM3 was arguably the most completely understood member of the CD225 family, but our finding that it acts as a SNARE assembly modulator significantly boosts our understanding of its key cellular functions. The closely related proteins, IFITM1, IFITM2, IFITM5, and IFITM10, are also likely to interact with SNARE proteins since they all share an R-SNARE-like motif (Fig. 1A). Interestingly, one of the most significant series of findings with regards to IFITM function was the characterization of a conserved amphipathic alpha helix in its amino terminus. This helix was shown to be essential for the antiviral activity of IFITM3 (Chesarino et al, 2017) and necessary for the ability of

IFITM3 to reduce cellular membrane fluidity (Rahman et al, 2020). Following our discovery here, we now know that this amphipathic helix is immediately adjacent to, or part of, an R-SNARE-like motif (Fig. 2A). Interestingly, a SNARE protein in yeast known as Spo20 also contains an amphipathic alpha helix amino terminal to its SNARE motif, resembling the structural organization of IFITM3 (Nakanishi et al, 2004). Moreover, another group of SNARE-binding proteins known as complexins are also bestowed with an amphipathic helix, and they use it to induce membrane curvature and stabilize fusion pores produced by SNARE complexes (Courtney et al, 2022). Therefore, it is possible that IFITM3 employs its amphipathic helix in conjunction with its R-SNARE-like motif to sculpt membranes at precisely the sites where endolysosomal SNARE assemblies are triggering membrane fusion. Other structural features lend additional support to our case that IFITM3 is a SNARE-like protein. Some SNARE proteins encode a single transmembrane domain to directly anchor into vesicular or target membranes (Jahn and Scheller, 2006), and IFITM3 is similar in this regard (Ling et al, 2016). In addition, palmitoylation is a post-translational modification of cysteine residues that promotes membrane localization, and IFITM3 and various

SNARE proteins are known to be palmitoylated (Fukasawa et al, 2004; Hach et al, 2013; Prekeris et al, 2000; Veit et al, 1996; Yount et al, 2012).

Notably, the relatively limited structural information previously available for IFITM3 did not indicate the presence of a continuous alpha helix reminiscent of a SNARE motif. In isolation, the region comprising the R-SNARE-like sequence motif of IFITM3 was previously thought to form one or two amphipathic helices followed by a disordered, intracellular loop (Chesarino et al, 2017; Ling et al, 2016). The structure of another CD225 protein, PRRT2, was posited to contain a similarly disordered, intracellular loop as well (Rossi et al, 2016). The typical SNARE fold characteristic of a SNARE motif may only form in the context of a coiled-coil assembly of multiple SNARE alpha helices (trans-SNARE complex). Our description of IFITM3 as a SNARE-like protein capable of interacting with endosomal SNARE proteins suggests that multiple protein folds and membrane topologies of IFITM3 may be possible, depending on the presence of interacting partners (such as STX7). Thus, attempts to structurally characterize IFITM3 in complex with STX7 or other endosomal SNAREs may provide substantial structural insight that has been lacking.

Regarding the structural interface between IFITM3 and endosomal SNARE proteins, the stoichiometry of the interactions remains unclear. We previously characterized a GxxxG motif present in CD225 proteins that enables homomultimerization and which is essential to the functions of PRRT2 and IFITM3 (Rahman et al, 2020). Here we show that mutation of the GxxxG motif in PRRT2 (G305W), TUSC5 (G141W), or IFITM3 (G95L) caused loss of binding with Q-SNAREs SNAP-25, STX4, or STX7, respectively (Figs. 1B, EV1C,D and 3A). These glycines reside adjacent to the +3 hydrophobic layer (Fig. 1A) and may disrupt the structural integrity of the R-SNARE-like motif. Alternatively, these findings may signal a requirement for CD225 proteins to multimerize prior to gaining SNARE binding potential. Another possibility is that the GxxxG motif in CD225 proteins not only controls homomultimerization but also impacts binding with other proteins including SNAREs.

What is more certain is that this previously unrecognized ability of IFITM3 to bind endosomal SNARE proteins and inhibit homotypic late endosome fusion advances our understanding of how IFITM3 inhibits the cellular entry of a diverse spectrum of enveloped and non-enveloped viruses (Majdoul and Compton, 2021). By showing that IFITM3 inhibits homotypic late endosome fusion while reinforcing heterotypic late endosome fusion, whereby late endosomes fuse with lysosomes, we provide an explanation for how IFITM3 overexpression causes expansion of the endolysosomal compartment and degradation of virions within it (Feeley et al, 2011). Taken together with previous findings, we propose a revised model for the molecular mechanism of IFITM3-mediated antiviral activity that is two-pronged: (1) IFITM3 traps incoming virions in late endosomes by inhibiting membrane fusion between the viral and cellular lipid bilayers; and (2) IFITM3 promotes the fusion of late endosomes with lysosomes to facilitate the degradation of trapped viral cargo.

In addition to blocking infection through inhibition of virus entry into cells, IFITM3 also performs an antiviral function in cells that are already infected—it has been shown to interact with and inhibit the function of viral fusogens (known as viral envelope glycoproteins) found in virions, which in turn reduces the infectivity of said virions (Compton et al, 2014; Tartour et al, 2014; Wang et al, 2017; Yu et al, 2015). Remarkably, viral fusogens

share structural and functional similarities to SNARE proteins, including heptad repeats and the formation of coiled-coil assemblies to drive membrane merger (Skehel and Wiley, 1998; Sollner, 2004). Therefore, our description of an R-SNARE-like motif in IFITM3 may reveal the molecular basis by which IFITM3 interacts with both cellular and viral fusion machinery to control virus infections at multiple levels.

# Methods

**Reagents and tools table**

| Reagent/Resource | Reference or Source | Identifier or Catalog Number |
|---|---|---|
| **Experimental models** | | |
| HEK293T | ATCC | CRL-3216 |
| HeLa | ATCC | CCL-2 |
| HeLa IFITM3 KO | Shi et al, 2018, PNAS; PMID: 30301809 | Dr. Alex A Compton, NCI |
| **Recombinant DNA** | | |
| pET3XC | Datta et al, 2009, Methods Mol Biol; PMID: 19020827 | Dr. Alan Rein, NCI |
| pQCXIP-Empty Vector | Compton et al, 2014, Cell Host Microbe; PMID: 25464829 | Dr. Alex A Compton, NCI |
| pQCXIP-FLAG-IFITM3 WT | Compton et al, 2014, Cell Host Microbe; PMID: 25464829 | Dr. Alex A Compton, NCI |
| pQCXIP-FLAG-IFITM3 G95L | Rahman et al, 2020, eLife; PMID: 33112230 | Dr. Alex A Compton, NCI |
| pQCXIP-FLAG-IFITM3 F75/78A | This study | N/A |
| pQCXIP-FLAG-IFITM3 R85Q | This study | N/A |
| pQCXIP-PRRT2-FLAG WT | This study | N/A |
| pQCXIP-PRRT2-FLAG R295Q | This study | N/A |
| pQCXIP-PRRT2-FLAG G305W | This study | N/A |
| pQCXIP-PRRT2-FLAG A320V | This study | N/A |
| pQCXIP-FLAG-TUSC5-WT | This study | N/A |
| pQCXIP-FLAG-TUSC5-G141W | This study | N/A |
| pcDNA3.1-HA-SNAP25 | This study | N/A |
| pcDNA3.1-HA-STX1A | This study | N/A |
| pcDNA3.1-HA-STX4 | This study | N/A |
| pcDNA3.1-HA-STX7 | This study | N/A |
| pcDNA3.1-HA-STX8 | This study | N/A |
| pcDNA3.1-HA-Vti1b | This study | N/A |
| pcDNA3.1-HA-VAMP8 | This study | N/A |
| pcDNA3.1-Myc-VAMP8 | This study | N/A |
| pcDNA3.1-Myc-VAMP7 | This study | N/A |

 

| Reagent/Resource | Reference or Source | Identifier or Catalog Number |
|---|---|---|
| pMRXIP-GFP-STX7 | Addgene | 45921 |
| pmCherry-N1-VAMP8 | Addgene | 92424 |
| pEGFP-N3-LAMP1 | Addgene | 34831 |
| pmRFP-C3-Rab5 | Addgene | 14437 |
| pmCherry-Rab7a-7 | Addgene | 55127 |
| **Antibodies** | | |
| Anti-NP | InvivoMab | BE0159 |
| Anti-NP | Abcam | ab20343 |
| Anti-STX8 | Synaptic Systems | 110-083 |
| Anti-STX7 | Synaptic Systems | 110-072 |
| Anti-Vti1b | Synaptic Systems | 164-002 |
| Anti-VAMP8 | Synaptic Systems | 104-302 |
| Anti-VAMP7 | Abcam | ab36195 |
| Anti-FLAG M2 | Sigma | F1804 |
| Anti-HA | Biolegend | 901514 |
| Anti-HA | Abcam | ab9110 |
| Anti-Myc | Sigma | C3956 |
| Anti-IFITM3 | Abcam | ab109429 |
| Anti-IFITM2/3 | Proteintech | 66081-1-Ig |
| Anti-Actin | Santa Cruz Biotechnology | SC-47778 |
| Anti-Tubulin | Santa Cruz Biotechnology | SC-5286 |
| Goat anti-mouse DyLight 680/800 | Li-COR | 926-68070/926-32210 |
| Goat anti-rabbit DyLight 680/800 | Li-COR | 926-68071/926-32211 |
| Goat anti-mouse IgG (H + L) Alexa Fluor 488 | Invitrogen | A-11001 |
| **Oligonucleotides and other sequence-based reagents** | | |
| None | | |
| **Chemicals, Enzymes and other reagents** | | |
| Rosetta 2 DE3 pLysS Competent *E. coli* | Sigma | 71403-3 |
| 2-YT Medium | Sigma | Y1003 |
| DMEM | Gibco | 11965092 |
| IPTG | Tecknova | I3430 |
| Lysozyme | Sigma | L6876 |
| PMSF | Sigma | 10837091001 |
| Triton X-100 | Sigma | X100 |
| Complete Protease Inhibitor cocktail | Roche | 11697498001 |
| Benzonase | Sigma | 1016950001 |
| Anapoe-X-100 | Anatrace | APX100 |
| DTT | Gold Bio | DTT |
| AEBSF | Santa Cruz Biotechnology | SC-202041 |
| Sarkosyl | Sigma | 61743 |
| Strep-Tactin Sepharose Resin | IBA Lifesciences | 2-1201-002 |
| Bio-Rad Protein Assay | Bio-Rad | 5000001 |

| Reagent/Resource | Reference or Source | Identifier or Catalog Number |
|---|---|---|
| Superose 12 10/300 GL column | Cytiva | 29036225 |
| His-tagged recombinant STX7 | Innovative Research | IHUSTX7RN6HISLY50UG |
| Dynabeads Protein A Immunoprecipitation Kit | Invitrogen | 10006D |
| NuPAGE LDS Sample Buffer | Invitrogen | NP0007 |
| NuPAGE Sample Reducing Agent | Invitrogen | NP0004 |
| NuPAGE MES SDS Running Buffer | Invitrogen | NP0002 |
| Criterion XT 12% Bis-Tris polyacrylamide gel | Bio-Rad | 3450117 |
| Influenza A Virus H1N1 PR8 | Charles River Laboratories | A/PR/8/34 |
| μ-Slide 8 Well | Ibidi | 80826 |
| μ-Dish 35 mm | Ibidi | 80136 |
| Dextran Alexa Fluor 488 (10,000 MW) | Invitrogen | D22910 |
| FluoroBrite DMEM | Gibco | A1896701 |
| Fetal Bovine Serum | Hyclone | SH30396.03 |
| Magic Red Cathepsin Kit | Abcam | ab270772 |
| Lysotracker Deep Red | Invitrogen | L12492 |
| Intercept Antibody Diluent | Li-COR | 927-65001 |
| Cytofix/Cytoperm | BD | 554714 |
| PermWash Buffer | BD | 554723 |
| D-PBS | Gibco | 14190144 |
| PureCol | Advanced Biomatrix | 5005 |
| Hoechst 33342 | Invitrogen | H3570 |
| **Software** | | |
| Fiji | ImageJ | imagej.nih.gov/index.html |
| Prism | Graphpad | www.graphpad.com |
| ImageStudioLite | Li-COR | www.licor.com/bio/image-studio |
| **Other** | | |
| LSRFortessa | BD | |
| Odyssey CLx | Li-COR | |

## Tissue culture, cell lines, and expression constructs

HEK293T (CRL-3216) and HeLa (CCL-2) were purchased from ATCC and cultivated at 37 °C and 5% $CO_2$ in DMEM (Gibco) complemented with 10% fetal bovine serum (Hyclone) and 1% penicillin-streptomycin (Gibco). HeLa *IFITM3* KO cells were generated as previously described (Shi et al, 2018). pQCXIP retroviral vectors encoding human IFITM3 (tagged with FLAG on amino terminus), PRRT2 (tagged with FLAG on carboxy terminus), or TUSC5 (tagged with FLAG on the amino terminus) were previously described (Rahman et al, 2020) or synthesized by

Integrated DNA Technologies (IDT) as gBlock fragments and cloned into pQCXIP using BamH1 and EcoR1 restriction enzymes. Mutations were introduced via QuikChange Lightning Site-Directed Mutagenesis (Agilent). HEK293T stably expressing pQCXIP-Empty Vector, pQCXIP-FLAG-IFITM3 WT, or pQCXIP-FLAG-IFITM3 G95L, encoding amino-terminal FLAG tag, were described previously (Rahman et al, 2020). pcDNA3.1 encoding SNARE proteins (SNAP25, STX1A, STX4, STX7, STX8, Vti1b, VAMP8, and VAMP7) tagged with HA or Myc at the amino terminus were synthesized by IDT as gBlock fragments and cloned into pcDNA3.1 using BamH1 and EcoR1 restriction enzymes. Transfections were performed with Lipofectamine 2000 (Invitrogen). All cell lines tested negative for mycoplasma.

## Antibodies, immunoblotting, and immunoprecipitations

Whole cell lysis was performed in a buffer containing 50 mM Tris-HCl pH 7.4, 1.0% Triton X-100 (Sigma), 150 mM NaCl, and 1 mM EDTA. Immunoprecipitation was performed as follows: antibodies were loaded with Dynabeads (Invitrogen) for 1 h at 4 °C while rotating; Dynabead-conjugated antibodies were added to whole cell lysates for 3 h at 4 °C while rotating to capture antibody-antigen complexes; immunoprecipitated fractions were isolated using the DynaMag-2 magnet (Invitrogen) and washed three times with lysis buffer prior to addition of NuPage LDS Sample Buffer (Invitrogen) and NuPage Sample Reducing Agent (Invitrogen); the immunoprecipitated and non-immunoprecipitated fractions were heat denatured at 100 °C for 5 min; samples were loaded into Criterion XT 12% Bis-Tris polyacrylamide gels (Bio-Rad) for SDS-PAGE using NuPAGE MES SDS Running Buffer (Invitrogen). Proteins were transferred to Amersham Protran Nitrocellulose Membrane (GE Healthcare), and membranes were blocked with Intercept Blocking Buffer-PBS (Li-COR) prior to incubation with primary and secondary antibodies. The following antibodies were used in this study: anti-STX8 (110-083; Synaptic Systems); anti-STX7 (110-072; Synaptic Systems), anti-Vti1b (164-002; Synaptic Systems), anti-VAMP8 (104-302; Synaptic Systems), anti-VAMP7 (ab36195; Abcam), anti-HA (901514, clone 16B12; Biolegend), anti-HA (ab9110; Abcam (this was used for immunoprecipitation)), anti-Myc (C3956, Sigma), anti-FLAG M2 (F1804; Sigma), anti-IFITM3 (ab109429; Abcam), anti-IFITM2/3 (66081-1-Ig, Proteintech), anti-tubulin and anti-actin (SC-47778; Santa Cruz Biotechnology). Goat anti-mouse and goat anti-rabbit secondary antibodies conjugated to DyLight 680 or 800 (Li-COR) and the Li-COR Odyssey CLx imaging system were used for generating immunoblots. Membrane images were analyzed using ImageStudio-Lite (Li-COR).

## Sequence retrieval and protein alignments

Protein sequences for CD225 proteins, Q-SNAREs, and R-SNAREs were retrieved from UniProt. Multi-sequence alignments were performed with MEGA11 and manually trimmed to the 53-residue SNARE motif. UniProt numbers for SNAREs are as follows: syntaxin1, Q16623; syntaxin4, Q12846; syntaxin 6, O43752; syntaxin7, O15400; syntaxin8, Q9UNK0; SNAP23, O00161; Vti1a, Q96AJ9; Vti1b, Q9UEU0; VAMP1, P23763; VAMP2, P63027; VAMP3, Q15836; VAMP5, O95183; VAMP7, P51809; VAMP8, Q9BV40; Ykt6, O15498; Syn8p, P31377; Tomosyn/STXBP5, Q5T5C0; Amisyn/STXBP6, Q8NFX7.

## Recombinant IFITM3 protein production

Recombinant human IFITM3 protein (WT, F65/68A, and G95L) was produced by cloning the nucleotide sequence appended with a Strep-II tag at the amino terminus into pET3XC and transforming it into Rosetta 2 DE3 pLysS Competent Cells (Sigma). A single colony of transformed cells was grown in LB overnight at 37 °C. One mL of the culture was expanded in 200 mL 2YT medium (Sigma) at 37 °C until the optical density reached 0.4, and the temperature was lowered to 18 °C for 1 h before induction with 0.4 mM IPTG (Tecknova) for 20 h. Pelleted cells were frozen at −80 °C before proceeding to protein purification. Thawed cells were resuspended in ice cold lysis buffer (at 10% w/v) consisting of 20 mM Tris-HCl pH 7.6, 100 mM NaCl, 1 mM DTT, 0.5 mg/ml lysozyme (Sigma), 1 mM PMSF (Sigma), 0.025% (v/v) Triton X-100 (Sigma), and 1X Complete Protease Inhibitors (Roche). Sample was stirred at 4 °C for 20 min before applying 2 freeze-thaw cycles by transferring the sample container between dry ice and a room temperature water bath. The thawed lysate was stirred at room temperature for 15 min after adding 5 U/mL benzonase (Sigma) prior to pulse sonication for 30 s at 4 °C. Sample was clarified by centrifugation at $5000 \times g$ for 10 min and the supernatant was further centrifuged at $40,000 \times g$ for 60 min. The pellet was thoroughly resuspended in the lysis buffer (modified to include 1 mM EDTA and excluding lysozyme) and centrifuged at $40,000 \times g$ for 60 min. This process was then performed a second time. The pellet was extracted twice with 20 mM Tris-HCl pH 7.6, 100 mM NaCl, 20% glycerol, 2% (v/v) Anapoe-X-100 (Anatrace), 1 mM DTT (Gold Bio), and 0.1 mM AEBSF (Santa Cruz Biotechnology) at 4 °C while stirring for 4 h each. The protein in the pellet was extracted with 20 mM Tris-HCl pH 7.6, 0.6% Sarkosyl (w/v) (Sigma), and 2 mM DTT. Strep-II-fused IFITM3 was purified by affinity chromatography using Strep-Tactin Sepharose Resin (IBA Lifesciences). Protein was >90% pure as determined by SDS-PAGE and Coomassie blue staining. Protein concentration was determined by Bio-Rad Protein Assay and stored at a concentration of 15–30 mg/mL at −80 °C. In the final reaction mixture of purified, recombinant IFITM3, the Sarkosyl concentration was 0.00002%.

## Size exclusion chromatography of recombinant IFITM3

Size exclusion chromatography was performed on a Superose 12 10/300 GL column (GE Healthcare). The column was equilibrated with a buffer consisting of 20 mM Tris-HCl pH 8.0, 120 mM NaCl, 0.2% (v/v) Triton X-100, and 1 mM TCEP. 150 µg recombinant IFITM3 (WT, F75/78A, or G95L) was diluted into 300 µL of buffer and injected into the column. One mL fractions were collected and subjected to SDS-PAGE and Coomassie blue staining.

## In vitro protein–protein co-immunoprecipitation

Two µg of recombinant Strep-II tagged IFITM3 WT or F75/78A or G95L were incubated with 2 µg of recombinant His-tagged STX7 (IHUSTX7RN6HISLY50UG; Innovative Research) in 0.3 mL of binding buffer (20 mM HEPES pH 7.4, 150 mM NaCl, 0.2% Triton X-100, and 0.1% BSA) overnight at 4 °C. The samples were incubated with 1 µg of anti-STX7 antibody for 1 h. The antibody-antigen complex was captured with 15 µL of Dynabeads (Invitrogen). After washing the samples 3 times with binding buffer (lacking BSA), proteins captured by beads using the DynaMag-2 were solubilized in 20 µL of 1X NuPAGE LDS Sample Buffer and

 

1X NuPAGE Sample Reducing Reagent (Invitrogen), and heat denatured at 100 °C for 5 min. SDS-PAGE and immunoblotting were performed and proteins were detected as described above.

## Structural modeling of the R-SNARE-like motif of IFITM3

The R-SNARE-like motif of IFITM3 (residues 55–113) and the R-SNARE motif of VAMP8 (residues 13–65) were compared by predictive protein structure alignment and comparison algorithm TM-align (Zhang and Skolnick, 2005). A TM score of 0.75847 was recorded, indicating that the two regions are likely to adopt the same alpha helical protein fold. Structural comparison between the R-SNARE-like motif of IFITM3 and the R-SNARE motif of VAMP8, which was previously crystallized as part of the STX7-STX8-Vti1b-VAMP8 trans-SNARE complex (PDB: 1GL2) (Antonin et al, 2002), was performed using Alphafold version 3 (Jumper and Hassabis, 2022) with assistance from the Colabfold server (Mirdita et al, 2022). A structure for the R-SNARE-like motif of IFITM3 was predicted using the R-SNARE motif of VAMP8 as a template. The Alphafold structure was then aligned with that of VAMP8 using the FATCAT server (Li et al, 2020). The aligned structures were then visualized and reoriented for figure presentation using ChimeraX (Meng et al, 2023).

## Influenza A Virus infection

Influenza A Virus [A/PR/8/34 (PR8), H1N1] supplied as clarified allantoic fluid was purchased from Charles River Laboratories. Infectious virus titers were calculated using a flow cytometry-based method in HEK293T cells (Grigorov et al, 2011), and infections were performed as follows: HEK293T cells (transiently transfected with 1.5 µg pQCXIP-Empty Vector, pQCXIP-FLAG-IFITM3 WT, or pQCXIP-FLAG-IFITM3 G95L were seeded in 24-well plates (50,000 per well) overnight and overlaid with virus diluted in 225 µL of complete DMEM (multiplicity of infection = 0.01) for ~18 h. Cells were washed with 1X D-PBS (Gibco), detached with Trypsin-EDTA (Gibco), fixed/permeabilized with Cytofix/Cytoperm (BD), immunostained with anti-IAV NP (ab20343; Abcam), and analyzed on a LSRFortessa flow cytometer (BD).

## Influenza A Virus cellular entry assay

An 8-well µ-Slide dish (80826; Ibidi) was coated with 3 mg/mL PureCol (Advanced Biomatrix) for 1 h at room temperature and then allowed to dry overnight. HEK293T stably expressing pQCXIP-Empty Vector, pQCXIP-FLAG-IFITM3 WT, or pQCXIP-FLAG-IFITM3 G95L were seeded at 20,000 cells per well in DMEM completed with 10% fetal bovine serum and 1% penicillin-streptomycin and incubated at 37 °C overnight. Cells were placed on ice for 5 min and then remained on ice while medium was removed and replaced with 120 µL complete DMEM containing Influenza A virus PR8 (multiplicity of infection = 1). Cells mock inoculated with PBS served as a negative control. Inoculated cells were incubated on ice for 40 min to allow for virus adhesion. Medium was removed, cells were washed twice with D-PBS, and medium was replaced with 150 µL complete DMEM containing 50 nM Lysotracker Deep Red (L12492; Invitrogen) and incubated at 37 °C for 4 h. Medium was removed, cells were washed twice with D-PBS, and cells were fixed/permeabilized with

Cytofix/Cytoperm (BD) for 10 min at room temperature. Cells were washed twice with PermWash buffer (BD) and blocked with Intercept Blocking Buffer (PBS) (Li-COR) for 1 h at room temperature. Cells were immunostained with anti-NP (BE0159; InvivoMab) at a 1:200 dilution in Intercept Antibody Diluent (PBS) (Li-COR) for 1 h at room temperature. Cells were washed twice in PermWash buffer and goat anti-mouse IgG (H + L) Alexa Fluor 488 secondary antibody (Invitrogen) at 1:300 dilution in Intercept Antibody Diluent (PBS) was added for 1 h at room temperature. Cells were washed twice in PermWash Buffer and Hoechst 33342 (H3570; Invitrogen) was added for 5 min at room temperature. Cells were washed once in PermWash Buffer and overlaid with 200 µL D-PBS for image acquisition on a Leica Stellaris 5 confocal microscope using the 63X oil objective. Confocal Z stacks of 20–30 optical sections were acquired for each field of view, and multiple fields were imaged per condition. Images were extracted and subjected to analysis in Fiji. To quantify Lysotracker staining, the corrected total cell fluorescence (CTCF) was measured for the middle Z slice of the entire field. To quantify nuclear NP accumulation, colocalization between NP and Hoechst was measured for the middle Z slick of each cell in the field and a ratio of NP+ nuclei: total nuclei was calculated.

## Late endosome-lysosome fusion assay with Dextran pulse-chase and Magic Red labeling

HEK293T cells stably expressing Empty Vector, IFITM3 WT-FLAG, or IFITM3 G95L-FLAG or HeLa cells (WT or *IFITM3* KO) were seeded in 35 mm low µ-Dishes (80136; Ibidi) coated with 3 mg/mL PureCol (Advanced Biomatrix) at 50,000 cells per well in DMEM completed with 10% fetal bovine serum and 1% penicillin-streptomycin and incubated at 37 °C overnight. Dextran Alexa Fluor 488 (10,000 MW) (D22910; Invitrogen) was added to cells at a concentration of 25 µg/mL for 2 h at 37 °C. Dextran-containing medium was then removed, cells were washed three times with D-PBS, fresh DMEM was added, and cells were incubated for 1 h at 37 °C. Cells were washed once with FluoroBrite DMEM (Gibco) completed with 10% fetal bovine serum and 1% penicillin-streptomycin and incubated with Magic Red cathepsin substrate (ab270772; Abcam) (from 250X stock in DMSO, a final concentration of 1X was used) for 5 min at 37 °C. Live cell image acquisition was performed on a Leica Stellaris 5 confocal microscope using the 63X oil objective.

## Confocal fluorescence microscopy

HEK293T cells stably expressing pQCXIP-Empty Vector, pQCXIP-FLAG-IFITM3 WT, or pQCXIP-FLAG-IFITM3 G95L or HeLa cells (WT or *IFITM3* KO) were seeded in DMEM completed with 10% fetal bovine serum and 1% penicillin-streptomycin at 500,000 cells per well in a 6-well plate and incubated at 37 °C overnight. Cells were transfected with 0.5 µg of the following plasmids: pMRXIP-GFP-STX7 (45921; Addgene), pmCherry-N1-VAMP8 (92424; Addgene), pEGFP-N3-LAMP1 (34831; Addgene), pmRFP-C3-Rab5 (14437; Addgene), or pmCherry-Rab7a-7 (55127; Addgene) using Lipofectamine 2000 (Invitrogen). Medium was replaced 6 h post-transfection, and cells were incubated at 37 °C for 18 h. Cells were washed once with D-PBS and fixed with Cytofix/Cytoperm (BD). For immunofluorescence

analysis of IFITM3 expression, immunostaining was performed with anti-FLAG M2 (F1804; Sigma) in fixed cells.

## EGFR-GFP degradation assay

HEK293T cells stably expressing pQCXIP-Empty Vector, pQCXIP-FLAG-IFITM3 WT, or pQCXIP-FLAG-IFITM3 G95L or HeLa cells (WT or *IFITM3* KO) were seeded in DMEM completed with 10% fetal bovine serum and 1% penicillin-streptomycin at 500,000 cells per well in a 6-well plate and incubated at 37 °C overnight. Cells were transfected with 0.5 μg of EGFR-GFP using Lipofectamine 2000. Medium was replaced 6 h post-transfection, and cells were incubated at 37 °C for 18 h. Recombinant EGF (R&D Systems) was added to cells and, at the indicate time point, cells were washed once with D-PBS and fixed with Cytofix/Cytoperm (BD). The percentage of EGFR-GFP+ cells was measured by flow cytometry on an LSRFortessa (BD).

## Statistics

Statistical differences were determined with one-way ANOVA test or student's t test corrected for multiple comparisons, using an alpha cut-off of 0.05. All tests were performed as two-tailed and unpaired. Test type and *P* values are reported in the figure captions.

# Data availability

This study includes no data deposited in external repositories. All data are available in the Source data files.

The source data of this paper are collected in the following database record: biostudies:S-SCDT-10_1038-S44318-024-00334-8.

# Peer review information

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

## Acknowledgements

We would like to thank Alexis Schirling for assisting in molecular cloning of SNARE proteins and Guoli Shi for assisting with confocal immunofluorescence microscopy. This work was supported by funding from the Intramural Research Program, National Institutes of Health, National Cancer Institute, Center for Cancer Research.

## Author contributions

**Kazi Rahman**: Conceptualization; Formal analysis; Investigation; Visualization; Writing—original draft; Writing—review and editing. **Isaiah Wilt**: Investigation; Writing—original draft; Writing—review and editing. **Abigail A Jolley**: Investigation; Writing—original draft. **Bhabadeb Chowdhury**: Investigation. **Siddhartha A K Datta**: Supervision; Investigation; Writing—original draft; Writing—review and editing. **Alex A Compton**: Conceptualization; Formal analysis; Supervision; Funding acquisition; Visualization; Writing—original draft; Writing—review and editing.

Source data underlying figure panels in this paper may have individual authorship assigned. Where available, figure panel/source data authorship is listed in the following database record: biostudies:S-SCDT-10_1038-S44318-024-00334-8.

## Disclosure and competing interests statement

The authors declare no competing interests.

# Expanded View Figures

**Figure EV1.  An R-SNARE-like motif is semi-conserved among human CD225 family members.**

(A) Human CD225 proteins were aligned and trimmed to a 53-residue stretch of sequence covering the SNARE motif. Proteins are grouped by whether they contain an arginine at the central "0" layer (R-SNARE-like) or not. Surrounding heptad repeats of hydrophobic residues are labeled from "−7" to "+8" in relation to the "0" layer, and sites highlighted in yellow correspond to hydrophobic residues. (B) Left: HEK293T cells were co-transfected with STX1A-HA and either PRRT2-FLAG (WT, R295Q, G305W, or A320V) or Empty Vector. SDS-PAGE and immunoblotting were performed with anti-HA and anti-FLAG in whole cell lysates. Anti-actin was used as loading control. Right: From co-transfected cells, STX1A-HA was immunoprecipitated with anti-HA followed by SDS-PAGE and immunoblotting with anti-HA and anti-FLAG. Immunoglobulin chain was used as loading control. (C) Left: HEK293T cells were co-transfected with STX4-HA and either TUSC5-FLAG (WT or G141W) or Empty Vector. SDS-PAGE and immunoblotting were performed with anti-HA and anti-FLAG in whole cell lysates. Anti-actin was used as loading control. Right: From co-transfected cells, TUSC5-FLAG was immunoprecipitated with anti-FLAG followed by SDS-PAGE and immunoblotting with anti-HA and anti-FLAG. Light chain immunoglobulin chain was used as loading control. The HA/FLAG ratio was calculated for the indicated lanes and shown as mean and standard error (normalized relative to WT, which was set to 100%). Differences that were statistically significant from WT as determined by student's T test are indicated by (*). Exact $p$ value: $p < 0.0001$. (D) From co-transfected cells, STX4-HA was immunoprecipitated with anti-HA followed by SDS-PAGE and immunoblotting with anti-FLAG and anti-HA. Light chain immunoglobulin was used as loading control. Numbers and tick marks left of blots indicate position and size (in kilodaltons) of protein standard in ladder. Immunoblots were performed independently three times, and a representative example is shown (except for (D), which was performed twice). Ig immunoglobulin, IP immunoprecipitation, EV Empty Vector, WT wild-type. Source data are available online for this figure.

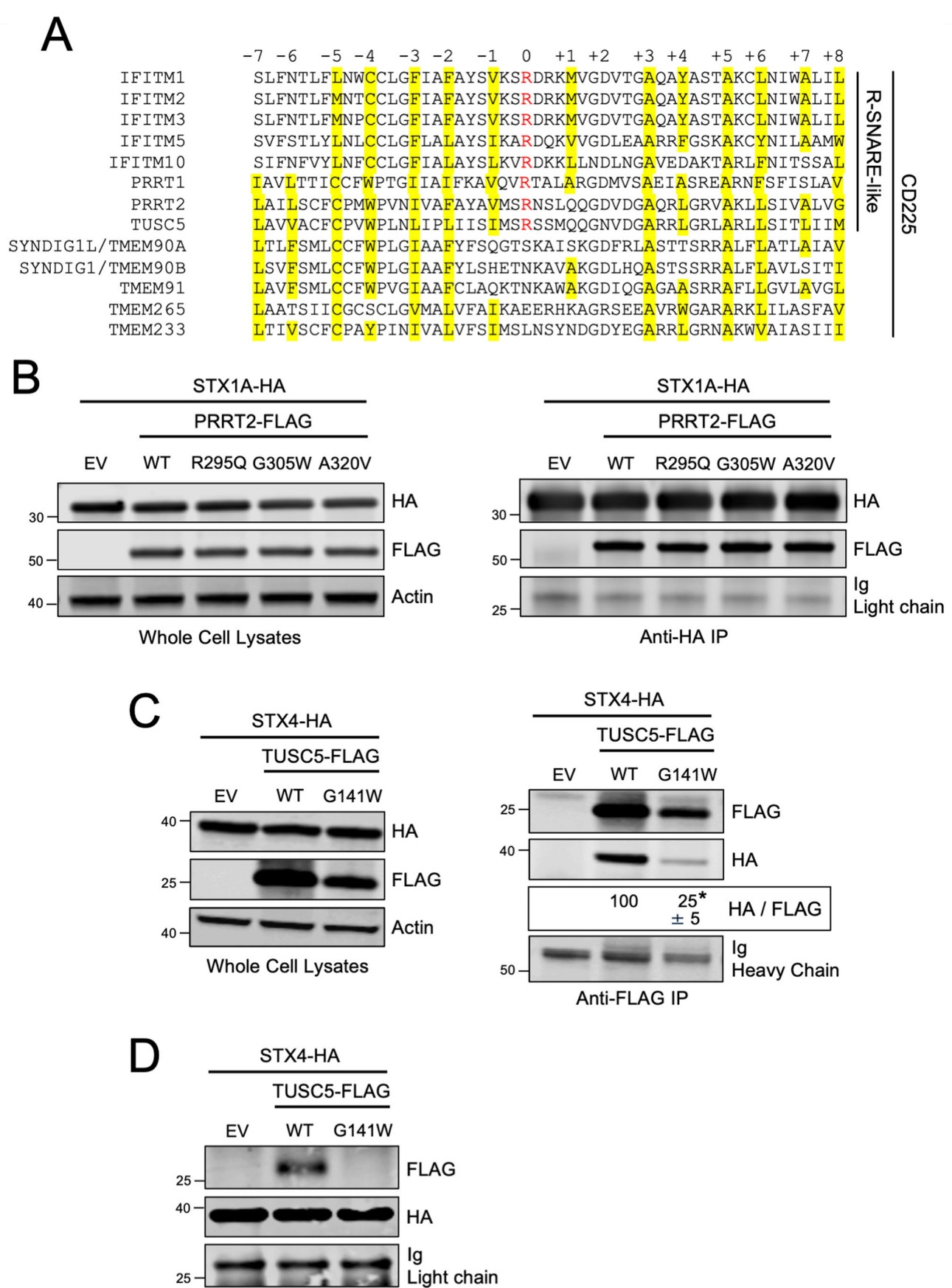

    

A

```
****************************************************************
*                    TM-align (Version 20190822)              *
* An algorithm for protein structure alignment and comparison *
* Based on statistics:                                        *
*       0.0 < TM-score < 0.30, random structural similarity   *
*       0.5 < TM-score < 1.00, in about the same fold         *
* Reference: Y Zhang and J Skolnick, Nucl Acids Res 33, 2302-9 (2005) *
* Please email your comments and suggestions to: zhng@umich.edu *
****************************************************************

Name of Chain_1: A540934
Name of Chain_2: B540934
Length of Chain_1:   53 residues
Length of Chain_2:   53 residues

Aligned length=   52, RMSD=   1.39, Seq_ID=n_identical/n_aligned= 0.019
TM-score= 0.75847 (if normalized by length of Chain_1)
TM-score= 0.75847 (if normalized by length of Chain_2)
```

B

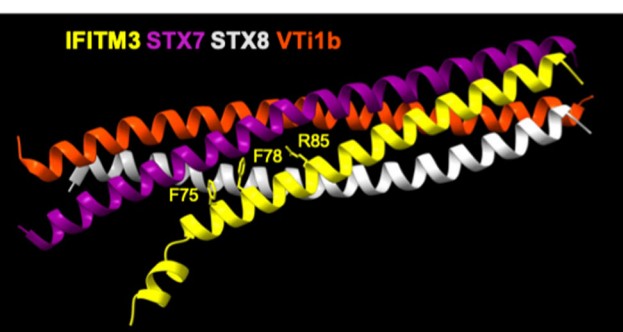

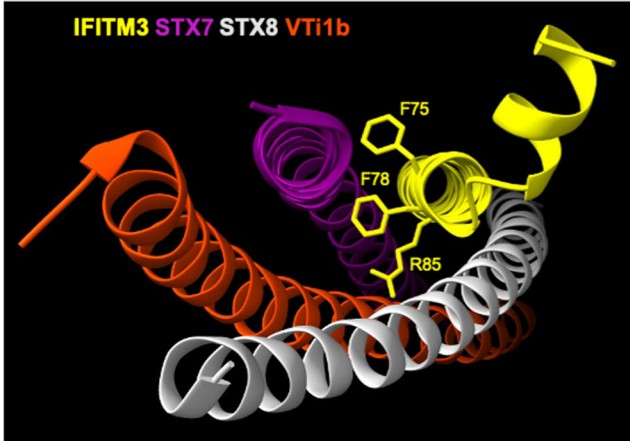

C

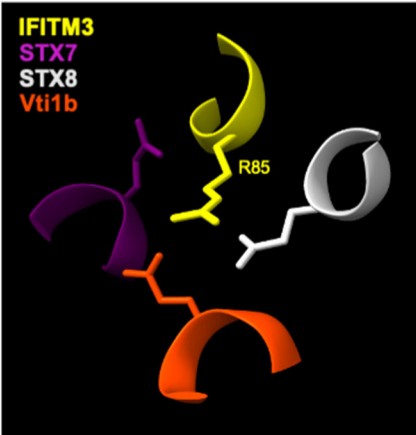

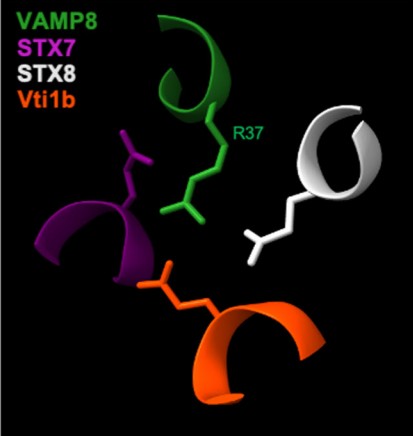

**Figure EV2.   IFITM3 contains a motif resembling the alpha helical R-SNARE motif of VAMP8.**

(A) The R-SNARE-like motif of IFITM3 and the R-SNARE motif of VAMP8 (53 residues each) were compared by predictive protein structure alignment and comparison algorithm TM-align. A TM score of 0.75847 was recorded, indicating that the two regions are likely to adopt the same alpha helical protein fold. (B) Structural prediction of the R-SNARE-like motif of IFITM3 was performed with Alphafold and FATCAT. Residues 55–113 of IFITM3 were modeled against a template of the R-SNARE motif of VAMP8 which was previously crystallized as part of the STX7-STX8-Vti1b-VAMP8 trans-SNARE complex (PDB: 1GL2). VAMP8 was then swapped with the predicted structure of IFITM3 and shown in relation to the coiled-coiled structure formed with STX7, STX8, and Vti1b. Top: the side chains of residues F75, F78, and R85 of IFITM3 are depicted in yellow and labeled. Bottom: alternative view of the side chains of F75, F78, and R85 of IFITM3. (C) Top: examination of the central polar "O" layer of the Q-SNAREs STX7, STX8, Vti1b in relation to the R85 residue in the predicted configuration of IFITM3. Bottom: examination of the central polar "O" layer of the STX7-STX8-Vti1b-VAMP8 complex (PDB: IGL2). Source data are available online for this figure.

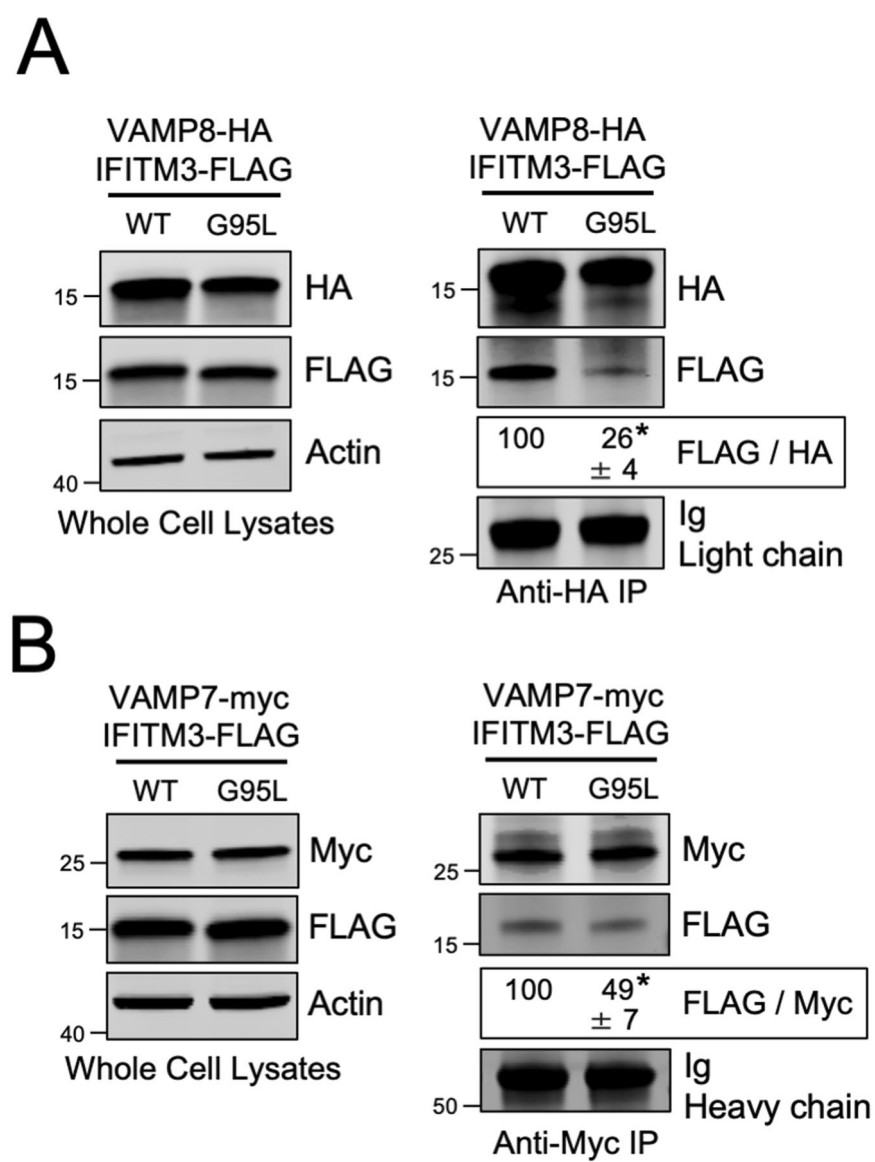

**Figure EV3. IFITM3 interacts with the R-SNAREs VAMP8 and VAMP7.**

(A) Left: HEK293T cells were co-transfected with VAMP8-HA and IFITM3-FLAG (WT or G95L). SDS-PAGE and immunoblotting were performed with anti-HA and anti-FLAG in whole cell lysates. Anti-actin was used as loading control. Right: From co-transfected cells, VAMP8-HA was immunoprecipitated with anti-HA followed by SDS-PAGE and immunoblotting with anti-HA and anti-FLAG. Light chain immunoglobulin chain was used as loading control. The FLAG/HA ratio was calculated for the indicated lanes and shown as mean and standard error (normalized relative to WT, which was set to 100%). Differences that were statistically significant from WT as determined by student's T test are indicated by (*). Exact p value: $p < 0.0001$. (B) Left: HEK293T cells were co-transfected with VAMP7-myc and IFITM3-FLAG (WT or G95L). SDS-PAGE and immunoblotting were performed with anti-myc and anti-FLAG in whole cell lysates. Anti-actin was used as loading control. Right: From co-transfected cells, VAMP7-myc was immunoprecipitated with anti-myc followed by SDS-PAGE and immunoblotting with anti-myc and anti-FLAG. Light chain immunoglobulin chain was used as loading control. The FLAG/Myc ratio was calculated for the indicated lanes and shown as mean and standard error (normalized relative to WT, which was set to 100%). Differences that were statistically significant from WT as determined by student's T test are indicated by (*). Exact p value: $p = 0.0023$. Numbers and tick marks left of blots indicate position and size (in kilodaltons) of protein standard in ladder. Immunoblots were performed independently three times, and a representative example is shown. Ig immunoglobulin, IP immunoprecipitation, WT wild-type. Source data are available online for this figure.

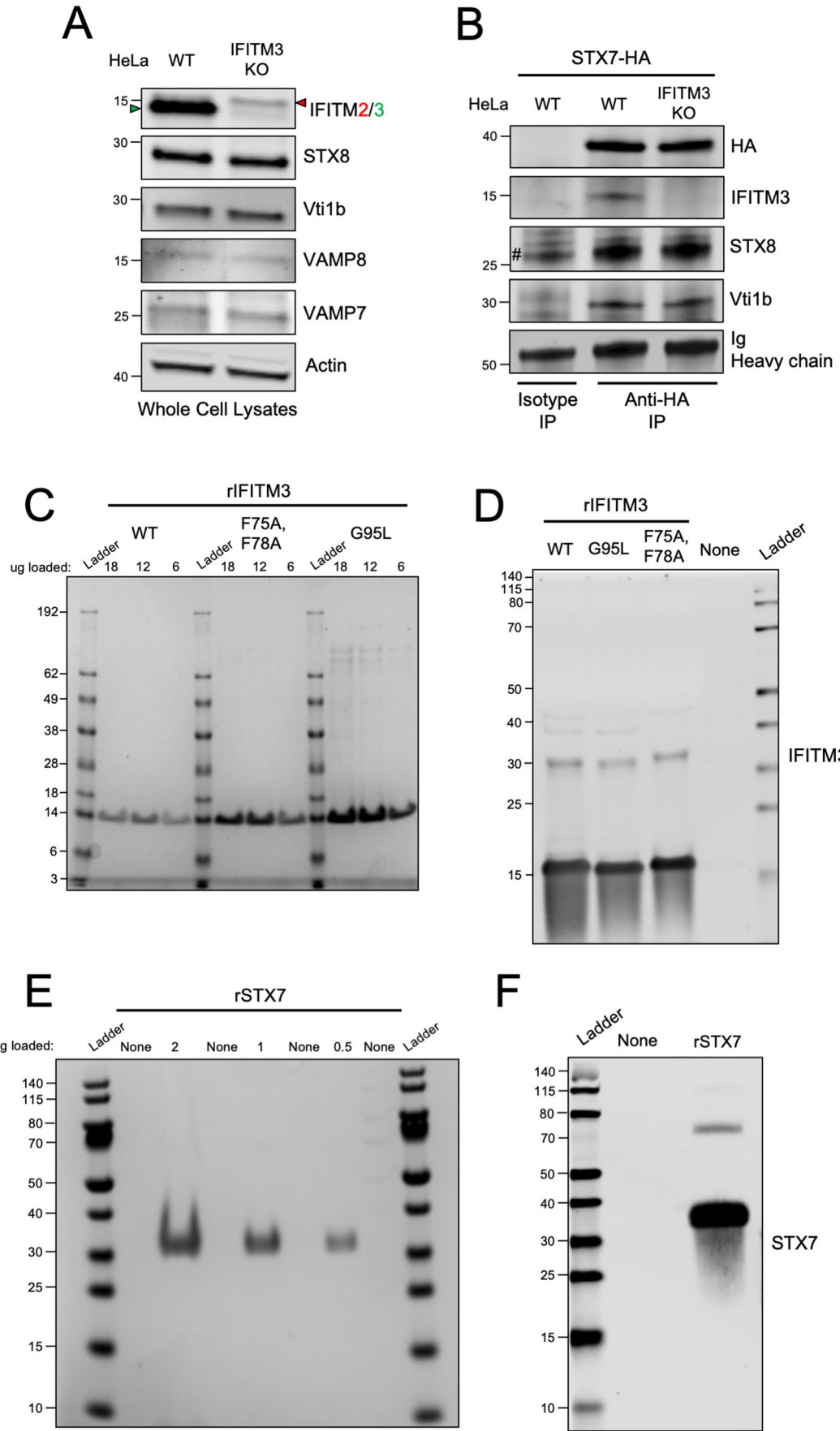

◀ **Figure EV4. IFITM3 does not impair the STX7/STX8 or STX7/Vti1b interactions.**

(A) HeLa cells (WT or *IFITM3* KO) were subjected to whole cell lysis, SDS-PAGE, and immunoblotting with anti-IFITM2/3, anti-STX8, anti-Vti1b, anti-VAMP8, anti-VAMP7, and anti-actin. Red and green arrows indicate IFITM3 and IFITM2, respectively, recognized by the anti-IFITM2/3 antibody. (B) STX7-HA was transfected into HeLa (WT or *IFITM3* KO) and immunoprecipitated with anti-HA followed by SDS-PAGE and immunoblotting with anti-HA, anti-IFITM3, anti-STX8, and anti-Vti1b. Heavy chain immunoglobulin was used as loading control. An isotype matched antibody was used as a control for immunoprecipitation. The (#) symbol denotes the presence of light chain immunoglobulin. (C) Purified recombinant IFITM3 protein (WT, F75/78A, or G95L) of varying inputs (6, 12, or 18 μg) were subjected to SDS-PAGE and visualized by Coomassie stain. (D) 20 ng of purified recombinant IFITM3 protein (WT, F75/78A, or G95L) were subjected to SDS-PAGE and immunoblotting with anti-IFITM3. (E) Recombinant STX7 of varying inputs (0.5, 1, or 2 μg) were subjected to SDS-PAGE and visualized by Coomassie stain. (F) Recombinant STX7 was subjected to SDS-PAGE and immunoblotting with anti-STX7. Numbers and tick marks left of blots indicate position and size (in kilodaltons) of protein standard in ladder. Immunoblots were performed independently two times and a representative example is shown (for (A) and (B)) or once (for (C), (D), (E), and (F)). Ig immunoglobulin, IP immunoprecipitation, EV Empty Vector, WT wild-type, r recombinant. Source data are available online for this figure.

 

A

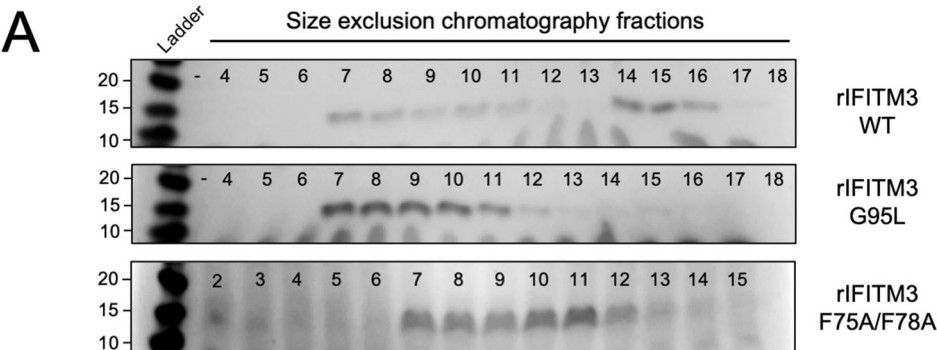

Size exclusion chromatography fractions

rIFITM3 WT

rIFITM3 G95L

rIFITM3 F75A/F78A

B

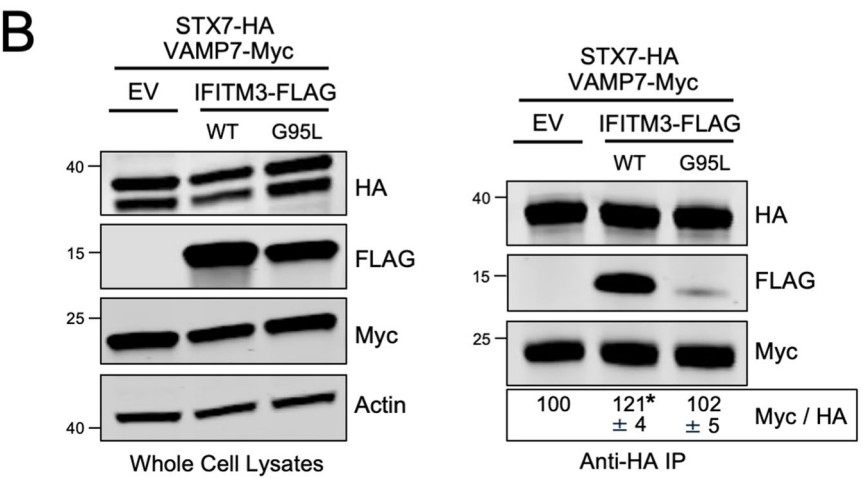

STX7-HA
VAMP7-Myc

Whole Cell Lysates

Anti-HA IP

C

STX7-GFP
VAMP8-mCherry

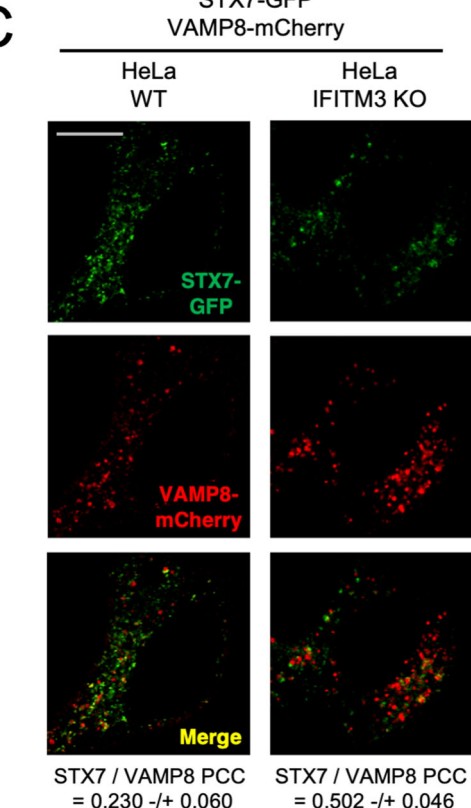

| HeLa WT | HeLa IFITM3 KO |
|---|---|
| STX7-GFP | |
| VAMP8-mCherry | |
| Merge | |
| STX7 / VAMP8 PCC = 0.230 -/+ 0.060 | STX7 / VAMP8 PCC = 0.502 -/+ 0.046 |

◀ **Figure EV5. IFITM3 does not impair the STX7/VAMP7 interaction.**

(A) Size exclusion chromatography was performed on recombinant IFITM3 proteins (WT, F75/78A, and G95L) and eluted fractions were analyzed by SDS-PAGE and Coomassie staining. 150 μg recombinant protein was loaded as input and fraction numbers correspond to mL volumes eluted from column. (B) Left: HEK293T cells stably expressing Empty Vector, IFITM3 WT-FLAG, or IFITM3 G95L-FLAG were co-transfected with STX7-HA and VAMP7-Myc. Whole cell lysates were subjected to SDS-PAGE and immunoblotting with anti-HA, anti-FLAG, and anti-Myc. Actin was used as a loading control. Right: STX7-HA was immunoprecipitated with anti-HA followed by SDS-PAGE and immunoblotting with anti-HA, anti-FLAG, and anti-Myc. The Myc/HA ratio was calculated for the indicated lanes and shown as mean and standard error (normalized relative to Empty Vector, which was set to 100%). Differences that were statistically significant from Empty Vector as determined by one-way ANOVA are indicated by (*). Exact p values are as follows (from left to right): $p = 0.0099$, $p = 0.8782$. Numbers and tick marks left of blots indicate position and size (in kilodaltons) of protein standard in ladder. Immunoblots were performed independently three times, and a representative example is shown (except (A), which was performed once). (C) HeLa (WT or *IFITM3* KO) were transfected with STX7-GFP and VAMP8-mCherry and confocal immunofluorescence microscopy was performed. Colocalization between STX7-GFP and VAMP8-mCherry was measured by calculating the Pearson's correlation coefficient using Fiji software. Coefficients were calculated from medial Z-slices from three fields of view containing 5–15 cells per condition and presented as means and standard error. Scale bar = 15 microns. IP immunoprecipitation, WT wild-type, r recombinant. Source data are available online for this figure.

 