## [Peer Review File · The EMBO Journal]

SNARE mimicry by the CD225 domain of IFITM3 enables regulation of homotypic late endosome fusion

Alex Compton, Kazi Rahman, Isaiah Wilt, Abigail Jolley, Bhabadeb Chowdhury, and Siddhartha Datta

Corresponding author: Alex Compton (alex.compton@nih.gov)

Review Timeline:

Submission Date:	16th Jul 24
Editorial Decision:	16th Aug 24
Revision Received:	12th Oct 24
Editorial Decision:	5th Nov 24
Revision Received:	11th Nov 24
Accepted:	22nd Nov 24

Editor: Ieva Gailite

Transaction Report:

Dear Dr. Compton,

Thank you for submitting your manuscript for consideration by the EMBO Journal. We have now received comments from three reviewers, which are included below for your information.

As you can see, all reviewers are generally positive in their assessment of the study, while also raising a number of concerns that would need to be addressed before they can support publication. From my side, I find the reviewers' requests generally reasonable. I would therefore invite you to address these comments in a revised manuscript. I think that it would be useful to discuss the revision in more detail via email or phone/videoconferencing - please let me know which option you prefer.

We generally allow three months as standard revision time, which can be extended to six months in the case of major revisions. Should you foresee a problem in meeting this deadline, please let us know in advance to discuss an extension.

As a matter of policy, competing manuscripts published during this period will not negatively impact on our assessment of the conceptual advance presented by your study. However, please contact me as soon as possible upon publication of any related work to discuss the appropriate course of action.

When preparing your letter of response to the referees' comments, please bear in mind that this will form part of the Review Process File and will therefore be available online to the community. For more details on our Transparent Editorial Process, please visit our website: <https://www.embopress.org/page/journal/14602075/authorguide#transparentprocess>. Please also see the attached instructions for further guidelines on preparation of the revised manuscript.

Please feel free to contact me if have any further questions regarding the revision. Thank you for the opportunity to consider your work for publication, and I look forward to discussing your revision with you.

With best regards,

Ieva

Ieva Gailite, PhD
Senior Scientific Editor
The EMBO Journal
Meyerohofstrasse 1
D-69117 Heidelberg
Tel: +4962218891309
i.gailite@embojournal.org

We realize that it is difficult to revise to a specific deadline. In the interest of protecting the conceptual advance provided by the work, we recommend a revision within 3 months (14th Nov 2024). Please discuss the revision progress ahead of this time with the editor if you require more time to complete the revisions.

Referee #1:

Humans encode for tree interferon-induced transmembrane proteins (IFITMs) with documented antiviral activity, IFITM1, IFITM2 and IFITM3. Expression of IFITM3 inhibits cellular entry of diverse enveloped viruses, including the Influenza A virus. However, the mechanism of virus restriction by IFITMs is not fully understood. A consensus model posits that IFITMs inhibit viral fusion by altering the properties of cell membranes.

This current study by Rahman and colleagues reports a novel function of IFITM3 that involves a selective inhibition of SNARE proteins that promote homotypic late endosome fusion, which favors virus degradation in lysosomes. The authors identified a novel SNARE-like motif in IFITMs that appears to adopt a VAMP-like (R-SNARE-like) structure and showed that IFITM3 directly and selectively interacts with the endosomal Q-SNARE protein syntaxin 7 (STX7) and inhibits VAMP8 binding to the endosomal SNARE complex. Mutations that compromise the IFITM3's ability to bind STX7 abrogate the antiviral activity against the Influenza A virus. These exciting findings reveal a multifaceted mechanism by which IFITM3 blocks entry of unrelated enveloped viruses by: (i) preventing viral fusion with late endosomes, and (ii) favoring the fusion of late endosomes with lysosomes and thereby promoting degradation of incoming virions.

This is an important and impactful study that advances our understanding of the functions of IFITMs. More broadly, the authors provide critical evidence that other members of the CD225 (dispanin) family also contain an R-SNARE-like motif and regulate Q-SNARE-mediated membrane fusion. The manuscript is very well written, and the conclusions are supported by rigorous experimental data.

I have only a few relatively minor concerns.

1. The authors to use a C-terminally FLAG-tagged IFITM3 for most of co-IP experiments, although this construct is not clearly described in Methods. There is concern that, due to the IFITM3 enrichment in late endosomes/lysosomes, the FLAG tag is, at least partially, cleaved off (e.g., PMC3820858). This suggests that immunoblotting for a C-terminally appended FLAG would detect only a subset of IFITM3-FLAG that is localized to non-degrading compartments. The authors should at least acknowledge this caveat.
2. Fig. 2E: WT is overexpressed relative to mutants. There is concern that this can result in a better apparent co-IP of WT with STX7.
3. Suppl. Fig. 3A: Given the cross-reactivity of anti-IFITM3 antibody with IFITM2, it is unclear how these two proteins were distinguished and what's the rationale behind the conclusion that the IFITM2 level was not unaffected. The figure appears to imply that the two proteins have slightly different mobilities, but the rationale for this assumption is not explained. Related to this point, do we know if a fraction of IFITM3 co-immunoprecipitated with STX7 is, in fact, IFITM2?
4. Fig. 4A: Why not include inactive IFITM3 mutants other than G95L to investigate the role of IFITM3 oligomerization in STX7 binding?
5. Fig. 4D: It is difficult to tell the size of the SNARE complex on the blot, as the band is smeared. Should not be this a defined 200kDa band, like the one on Supplemental Figure 4B?
6. Suppl. Fig. 4A: Why does STX7 run as 2 bands in this figure? Also, the lower band in the EV lane is cut off.
7. Fig. 5A: Does the G95L mutant promote IAV infection?
8. Line 369 of Discussion: implies that IFITM3 may have multiple topologies. Is this assertion supported by data? It is unclear how STX7 binding may suggest different IFITM3 topologies.

Referee #2:

General summary and opinion about the principle significance of the study, its questions and findings:

Rahman K et al, in the study entitled "SNARE mimicry by the CD225 domain of IFITM3 enables regulation of homotypic late endosome fusion" shows that IFITM3 has a CD225 domain which mimicks SNARE fusogenic and regulates endo-lysosomal fusion. IFITM3 is a well-studied and important protein. It is a small interferon-inducible transmembrane protein which prevents the entry of several viruses. Upon interferon treatment, IFITM3 is expressed and targeted into late endosomes. It has been shown that IFITM3 can regulate cholesterol homeostasis in late endosomes and several studies showed that IFITM3 blocks virus-mediated membrane fusion. In addition, Spence et al, 2019 showed that IFITM3 directly engages and shuttles incoming virus particles to lysosomes. This work further supports the model proposed by Spence et al, 2019 and provides a plausible molecular mechanism. Based on the results, authors suggest that IFITM3 has an inhibitory mechanism which is based on its ability to compete with syntaxin 7 (STX7) and reduces the assembly of STX7-STX8-Vti 1b-VAMP8 SNARE complex involved in homotypic late endosome fusion. This disruption promotes endosome-lysosome fusion and hence also degradation of viruses. Most of the data in the manuscript are based on immunoprecipitation experiments using cell lysates and Western Blot analysis. In addition, the authors used in vitro immunoprecipitation using recombinant proteins and confocal microscopy to study influenza A virus entry in the presence of IFITM3 and IFITM3 G95L and also included additional mutants in the study. Overall, the study is interesting and it provides further characterization of IFITM3 molecular biology. Considering that this study shows a novel finding that IFITM3 is able to interact with SNARE proteins, it will be important to a broad audience in molecular and cell biology as well as in virology. However, the study requires experimental validation of the presented data and additional experiments to prove that IFITM3 regulates homotypic endosomal fusion and endo-lysosomal fusion in cells. As it is presented now, authors show that IFITM3 can interact with SNARE proteins in vitro and in cells and this interaction is impaired by G95L mutation. In addition, the authors could show that G95L is not able to block influenza A virus entry. However, the experiment directly showing that IFITM3 regulates endolysosomal compartment fusion, as stated in the title, is missing. In addition, authors should provide WB quantification from all three replications and report standard deviations including significance tests.

Specific major concerns essential to be addressed to support the conclusions:

- 1) The data shows that IFITM3 blocks viral entry and that IFITM3 G95L is impaired in influenza A virus entry. However, additional experimental data are needed to demonstrate that IFITM3 modulates membrane fusion of endosomal compartments with lysosomal compartments. For this authors should take advantage of well-established early and endosomal (Rab5 and Rab7) and lysosomal markers (LAMP) fused to fluorescent proteins and measure their colocalization and organelle size distribution in HEK stable cell lines expressing IFITM3, IFITM3 G95L and also in HeLa WT and HeLa IFITM3 KO. This should confirm whether IFITM3 is regulating homotypic endosomal fusion. In the presence of IFITM3, lysosomal compartments but not endosomal compartments should be larger.
- 2) Colocalization of IFITM3 and IFITM3 G95L with SNX7 should be provided to support the results of immunoprecipitation experiments. The authors mention in line: 282 "To test the functional association between SNARE modulation and lysosomal targeting ..." However, this was not directly tested in Figure 5.
- 3) To prove that the IFITM3 enhances virus degradation in lysosomes, authors should provide quantitative data on lysosome cargo degradation. They could use established published protocols: Protocol for quantification of the lysosomal degradation of extracellular proteins into mammalian cells (10.1016/j.xpro.2021.100975). In addition, lysosomal degradation of EGFR could be compared in the presence of IFITM3 and IFITM3 G95L as mentioned on Line 397. One would expect that IFITM3 but not IFITM3 G95L will increase cargo degradation.
- 4) In addition, a control colocalization analysis of IFITM3 and IFITM3 G95L with endosomal, and lysosomal markers should be done to ensure that IFITM3 G95L targeting into endosomes is not impaired.
- 5) Most of the conclusions are based on WB analysis of bands. Even though the authors mention that WB was repeated three times, they report a single value below the WB and it is not explained if this is an average of 3 repetitions or just determined from a single WB. Are these biological or technical replicates using the same samples? How many times were the WB repeated and what is the standard deviation of the data? In some WB there is not only one band but two bands are visible and the second band is cropped away. Why not including the second band in the analysis? Uncropped WB images for all experiments and replications should be provided.
- 6) Discussion about the structure of the amphipathic helix (Line 361-367): IFITM3 Cys72 is palmitoylated and Cys72 palmitoylation was shown to be important for IFITM3 function. It is localized between 2 amphipathic helices which are modelled here as one helix. Was palmitoylation at Cys72 used in the AlphaFold3 model presented in Figure 2C? Considering that palmitoyl is embedded in the membrane, would the SNARE complex formation with IFITM3 be sterically possible?
- 7) IFITM3 is a transmembrane protein, gel filtration chromatography elution profile should be provided to show that recombinant IFITM3 was not aggregated after affinity purification.

Minor concerns that should be addressed:

- The abstract should be restructured and shortened, the focus of the study is not immediately clear from the abstract.
- IFITM3 should be better introduced in the introduction including more details on the mechanism of action
- Line 222: Please clarify what "HeLa cells competent for IFITM3" mean. Were these cells overexpressing IFITM3 or was IFITM3 induced by interferon treatment?
- Based on the proposed mechanism, how would IFITM1 localized in plasma membrane function or does interaction with SNARE complex only apply to IFITM3 and IFITM2? For example, a previous study (Kun Li et al, 2013) showed that IFITM1 can arrest cell-cell fusion.
- Overall the manuscript would be improved by including a model as a figure on the proposed molecular mechanism

- Terms like "fresh insight", "newfound", and "leap" should be avoided.

Referee #3:

Rahman et al. show that members of the CD225 protein family harbor a putative R-SNARE motif, which interacts with Q-SNAREs and modulates SNARE complex assembly. Importantly, the work shows that mutations within this SNARE-like motif result in reduced SNARE binding and affect protein function. Many of these mutations have been previously associated with respective diseases. Beside some basic characterization of the putative SNARE motifs of PRRT2 and TUSC5, an in depth analysis of IFITM3 indicates that its SNARE-like motif contributes to the redirection of intracellular virus trafficking towards lysosomal degradation. Overall, the identification of SNARE motif-like sequences within the CD225 protein family profoundly forwards our understanding of the regulation of membrane trafficking and suggests that distinct CD225 family members may regulate different SNARE interactions.

The data are overall compelling and the paper is well written. Nevertheless, the manuscript could be improved by providing more detailed insights into the properties (SNARE-specific interactions) of the putative R-SNARE motif of IFITM3. The authors suggest that the IFITM3 mainly binds the Q-SNAREs STX7. Indeed, ectopic protein expression in HEK293 cells (Fig. 2C and supplemental Fig. 2C) and purified proteins (Fig. 3C) confirm an IFITM3 - STX7 interaction. However, instead of using a Western blot analysis (Fig. 3C), the authors should consider to repeat the pull downs with increasing concentrations of the IFITM3 variants followed by an analysis using SDS-PAGE and Coomassie blue staining (incl. quantification). (Technically, the authors could also use the protein purification tag for the pull down instead of the anti-STX7 antibody.) In case of saturable binding, such an experiment will provide a rough estimate about the binding affinities of IFITM3 variants. If possible, such binding experiments should also be performed with recombinant VAMP7 and VAMP8 (e.g. containing an affinity tag). Please note that CD225 family members also show direct v-SNARE interactions. PRRT2 directly interacts with VAMP2 (Coleman et al., 2018, Cell Reports) and a recent screen for IFITM3 binding partners identified VAMP7 as a potential partner (Li et al., 2024, International Journal of Biological Macromolecules). Importantly, potential differential v-SNARE - IFITM3 interactions may also provide mechanistic insights of how IFITM3 redirects viral trafficking. The cellular pull down results shown in Fig. 2D are not conclusive for VAMP7 and VAMP8 due to vastly different expression levels. (Reverse co-immunoprecipitation experiments were not performed.) Actually, the data shown in supplemental Fig. 4A and Fig. 4C would support a positive effect of IFITM3 for VAMP7 interactions.

Additional points:

- Most of the Western blot analysis, showing protein interacts, contain quantifications. It seems that these pull down experiments were performed three times. This should be reflected in the quantifications, better illustrating reproducibility.
- Figure 1B: To firmly exclude that different expression levels of the PRRT2 variants cause the reduced binding of the PRRT2 mutants in the pull down with SNAP-25-HA, the authors should consider to show a shorter exposure of anti-FLAG signal in the whole lysate. (As a note, in supplemental Fig. 1, all PRRT2 constructs show similar expressions levels and the mutants do not affect STX1A binding.) In this context, please also show a shorter exposure of the TUSC5 levels in the whole lysate (supplemental Fig. 1C). Can the authors provide some information about the pull down efficiencies?
- In Figure 2C, assuming that IFITM3 indeed functions as an R-SNARE, the authors may consider to include a potential model of STX7/STX8/Vti1b/IFITM3 complex, also showing the positions of the tested mutants and how these mutants may affect/interfere with IFITM3-SNARE complex formation/interaction.
- In supplemental Fig. 2C, the F75/78A mutant, compared to the IFITM3 WT, shows enhanced binding to Vti1b (despite a lower expression level). Please provide an explanation.
- In Fig. 3B, supplemental Fig. 3B, and Fig 4B, it seems that the isotopic antibody precipitate STX8. Please clarify.
- In Materials and Methods, it is mentioned that supplemental Fig. 3D shows Coomassie blue staining, but the corresponding figure legends mentions immunoblotting with anti IFITM3 antibody. Please clarify. Why do the authors show an immunoblot, instead of a Coomassie blue stain of purified STX7 in supplemental Fig. 3E?
- In the Western blot analysis shown in Fig. 4D and supplemental Fig. 4B please also show the expression levels of the ectopic SNAREs (protein bands corresponding to MW of monomeric state). Additional Western blot analysis of the SDS-resistant bands (Fig. 4D) using anti-VAMP7 and anti-IFITM3 (FLAG) antibodies could further strengthen the manuscript. (For example, IFITM3 may not be part of the SDS-resistant complex, but VAMP7 may be increased in the presence of IFITM3.)
- Supplemental Fig. 4, figure legend: please replace the typo "IIP" by "IP".
- In most Western blot analysis, the Ig chain seems to migrate at 35 kDa. In others, it migrates above 40 kDa (e.g. Fig. 3A and B). The IgG heavy and light chains usually migrate at 50 and 25 kDa, respectively. Please briefly explain the discrepancy.

Referee #1:

Humans encode for three interferon-induced transmembrane proteins (IFITMs) with documented antiviral activity, IFITM1, IFITM2 and IFITM3. Expression of IFITM3 inhibits cellular entry of diverse enveloped viruses, including the Influenza A virus. However, the mechanism of virus restriction by IFITMs is not fully understood. A consensus model posits that IFITMs inhibit viral fusion by altering the properties of cell membranes.

This current study by Rahman and colleagues reports a novel function of IFITM3 that involves a selective inhibition of SNARE proteins that promote homotypic late endosome fusion, which favors virus degradation in lysosomes. The authors identified a novel SNARE-like motif in IFITMs that appears to adopt a VAMP-like (R-SNARE-like) structure and showed that IFITM3 directly and selectively interacts with the endosomal Q-SNARE protein syntaxin 7 (STX7) and inhibits VAMP8 binding to the endosomal SNARE complex. Mutations that compromise the IFITM3's ability to bind STX7 abrogate the antiviral activity against the Influenza A virus. These exciting findings reveal a multifaceted mechanism by which IFITM3 blocks entry of unrelated enveloped viruses by: (i) preventing viral fusion with late endosomes, and (ii) favoring the fusion of late endosomes with lysosomes and thereby promoting degradation of incoming virions.

This is an important and impactful study that advances our understanding of the functions of IFITMs. More broadly, the authors provide critical evidence that other members of the CD225 (dispanin) family also contain an R-SNARE-like motif and regulate Q-SNARE-mediated membrane fusion. The manuscript is very well written, and the conclusions are supported by rigorous experimental data.

We thank the reviewer for their thorough assessment and for noting the significant impact of our findings.

I have only a few relatively minor concerns.

1. The authors to use a C-terminally FLAG-tagged IFITM3 for most of co-IP experiments, although this construct is not clearly described in Methods. There is concern that, due to the IFITM3 enrichment in late endosomes/lysosomes, the FLAG tag is, at least partially, cleaved off (e.g., PMC3820858). This suggests that immunoblotting for a C-terminally appended FLAG would detect only a subset of IFITM3-FLAG that is localized to non-degrading compartments. The authors should at least acknowledge this caveat.

We recognize the reviewer's concern regarding the C-terminal epitope tagging of IFITM3 constructs. However, our IFITM3 constructs contain an N-terminal FLAG tag. This was stated in the Materials and Methods section. At line 459, we state "pQCXIP retroviral vectors encoding human IFITM3 (tagged with FLAG on amino terminus)..." We chose to utilize N-terminally tagged IFITM3 for the precise reason outlined by the reviewer – C-terminal epitope tags have been shown to be cleaved off in lysosomes. Thank you.

2. Fig. 2E: WT is overexpressed relative to mutants. There is concern that this can result in a better apparent co-IP of WT with STX7.

Indeed, we noticed that mutations in IFITM3 within the R-SNARE-like motif resulted in slightly lower protein expression levels. That is precisely why we performed anti-FLAG immunoprecipitation in order to enrich for IFITM3 and minimize the differences in IFITM3 protein loaded into the gel. Furthermore, these results were confirmed with the reverse co-IP

(new Figure 3A). At line 208, we mention that “IFITM3 F75/78A and R85Q exhibited somewhat decreased steady-state levels relative to WT, possibly attesting to the importance of the R-SNARE-like motif in maintaining proper folding and stability of the IFITM3 polypeptide.” Lastly, since we showed that recombinant IFITM3 WT could pull down recombinant STX7 to a greater extent than recombinant IFITM3 F75/78A and G95L (new Figure 4F), we can rule out that different protein inputs (either in cells or in vitro) contribute to the decreased interaction between IFITM3 F75/78A and G95L and STX7.

3. Suppl. Fig. 3A: Given the cross-reactivity of anti-IFITM3 antibody with IFITM2, it is unclear how these two proteins were distinguished and what's the rationale behind the conclusion that the IFITM2 level was not unaffected. The figure appears to imply that the two proteins have slightly different mobilities, but the rationale for this assumption is not explained. Related to this point, do we know if a fraction of IFITM3 co-immunoprecipitated with STX7 is, in fact, IFITM2?

As the reviewer points out, the antibody used in Supplemental Figure 3A is indeed cross-reactive for both IFITM3 and IFITM2. This was intentional, since we wanted to establish that IFITM3 KO cells were deficient for IFITM3 but still retained IFITM2 expression. The results suggest that that is indeed the case. The rationale for the differential mobility of endogenous IFITM3 and IFITM2 in HeLa cells has already been established by previous publications, including one of our own (PMID: 30301809). In this same publication, we characterized the specificity of this anti-IFITM2/3 antibody by comparing it with specific anti-IFITM3 and anti-IFITM2 antibodies, in cells that were deficient for IFITM3, IFITM2, or both. Therefore, we can be confident that what we identify as IFITM3, and what we identify as IFITM2, are accurate. These results are now found in new Extended View Figure 4A.

4. Fig. 4A: Why not include inactive IFITM3 mutants other than G95L to investigate the role of IFITM3 oligomerization in STX7 binding?

We thank the reviewer for this question. However, the G95L mutation is the only mutation known to significantly disrupt the oligomerization of IFITM3 (as determined in our previous publication PMID: 33112230). As a result, it was not possible to investigate the role of IFITM3 oligomerization in STX7 binding using any additional mutations. Nonetheless, our results showing that F75/78A also disrupts binding to STX7 reveals that loss of IFITM3 oligomerization is not the main mechanism causing loss of STX7 binding, because in our previous publication (PMID: 33112230) we show that the F75/78A mutations do not affect IFITM3 oligomerization (while G95L inhibits oligomerization).

5. Fig. 4D: It is difficult to tell the size of the SNARE complex on the blot, as the band is smeared. Should not be this a defined 200kDa band, like the one on Supplemental Figure 4B? The reviewer brings up a good point. However, when resolving the SNARE complex using the anti-Myc antibody as we did in Figure 4D, we noticed that the complex does not run as a single, highly resolved band. Rather, the anti-Myc blot shows a slight smear, which may reflect that 1) the gel used is unable to tightly resolve high order complexes, and a different concentration of acrylamide may be necessary, and/or 2) there exist multiple SNARE complexes containing VAMP8, which may differ in terms of composition and size. Since the band resolved with the anti-STX8 antibody in Supplemental Figure 4B shows less smearing, we can only speculate that one or both of these explanations are responsible. However, it may also be the product of slightly different running times during the electrophoresis step. What we think is important is the fact that, regardless of exact complex size or smeariness, IFITM3 WT decreases assembly of this complex containing STX7, STX8, Vti1b, and VAMP8. The results discussed can now be found in new Figure 6A and Appendix Figure S1A.

6. Suppl. Fig. 4A: Why does STX7 run as 2 bands in this figure? Also, the lower band in the EV lane is cut off.

We are unsure why STX7 runs as two bands in Supplemental Figure 4A, but this was also the case in Figure 4A. However, only one band was observed when STX7 was immunoprecipitated with the anti-HA antibody, and since our results emphasize how IFITM3 WT interacts with this form of STX7, we believe that is what is important. These results can now be found in new Figure 5A and new Extended View Figure 5B.

7. Fig. 5A: Does the G95L mutant promote IAV infection?

This is an interesting question. When we examined IAV infection in cells at 18 hours post-inoculation using an anti-NP antibody and flow cytometry, IFITM3 WT but not G95L reduced infection considerably relative to an Empty Vector control (Figure 2B). In this setting, the levels of infection observed in cells expressing G95L were not higher than those observed in Empty Vector. However, when we examined the nuclear NP levels at soon after inoculation, we noticed that nuclear NP was increased in G95L cells relative to Empty Vector cells (new Figure 7). Therefore, G95L may slightly enhance IAV entry when examined shortly after inoculation, but this effect is not observed when examining total NP staining at later time points. Therefore, we do not believe that G95L significantly enhances IAV infection.

8. Line 369 of Discussion: implies that IFITM3 may have multiple topologies. Is this assertion supported by data? It is unclear how STX7 binding may suggest different IFITM3 topologies. This statement, that IFITM3 may adopt multiple topologies, is based on the previous suggestion that IFITM3 encodes a disordered cytoplasmic intracellular loop within its CD225 domain. Since we are now showing that the CD225 domain is predicted to adopt an extended alpha-helical fold in reminiscent of R-SNARE proteins, these results seem to be at odds with one another. Therefore, it is possible that SNARE binding provides order to the CD225 domain, transforming a once disordered region into an alpha helix characteristic of other R-SNARE proteins. Our findings suggest that efforts to structurally characterize IFITM3 should include its interaction partners, such as STX7, because the complex of proteins could reveal structural information which may be missed when IFITM3 is characterized in isolation.

Referee #2:

General summary and opinion about the principle significance of the study, its questions and findings:

Rahman K et al, in the study entitled "SNARE mimicry by the CD225 domain of IFITM3 enables regulation of homotypic late endosome fusion" shows that IFITM3 has a CD225 domain which mimicks SNARE fusogenic and regulates endo-lysosomal fusion. IFITM3 is a well-studied and important protein. It is a small interferon-inducible transmembrane protein which prevents the entry of several viruses. Upon interferon treatment, IFITM3 is expressed and targeted into late endosomes. It has been shown that IFITM3 can regulate cholesterol homeostasis in late endosomes and several studies showed that IFITM3 blocks virus-mediated membrane fusion. In addition, Spence et al, 2019 showed that IFITM3 directly engages and shuttles incoming virus particles to lysosomes. This work further supports the model proposed by Spence et al, 2019 and provides a plausible molecular mechanism. Based on the results, authors suggest that IFITM3 has an inhibitory mechanism which is based on its ability to compete with syntaxin 7 (STX7) and reduces the assembly of STX7-STX8-Vti 1b-VAMP8 SNARE complex involved in homotypic late endosome fusion. This disruption promotes endosome-lysosome

fusion and hence also degradation of viruses. Most of the data in the manuscript are based on immunoprecipitation experiments using cell lysates and Western Blot analysis. In addition, the authors used *in vitro* immunoprecipitation using recombinant proteins and confocal microscopy to study influenza A virus entry in the presence of IFITM3 and IFITM3 G95L and also included additional mutants in the study.

Overall, the study is interesting and it provides further characterization of IFITM3 molecular biology. Considering that this study shows a novel finding that IFITM3 is able to interact with SNARE proteins, it will be important to a broad audience in molecular and cell biology as well as in virology. However, the study requires experimental validation of the presented data and additional experiments to prove that IFITM3 regulates homotypic endosomal fusion and endo-lysosomal fusion in cells. As it is presented now, authors show that IFITM3 can interact with SNARE proteins *in vitro* and in cells and this interaction is impaired by G95L mutation. In addition, the authors could show that G95L is not able to block influenza A virus entry. However, the experiment directly showing that IFITM3 regulates endolysosomal compartment fusion, as stated in the title, is missing. In addition, authors should provide WB quantification from all three replications and report standard deviations including significance tests.

We thank the reviewer for their thorough and positive assessment of our work and for identifying its significance to a broad audience in multiple scientific fields. In response to the reviewer's criticism that we did not directly show that IFITM3 regulates endolysosomal fusion, we now include new data using an *in vivo* late endosome-lysosome fusion assay in living cells. This assay entails pulsing cells with Dextran, chasing the Dextran by exchanging medium, and measuring its colocalization with Magic Red, a substrate for cathepsin activity in lysosomes (see below). In response to the reviewer's other concern, we now provide western blot quantification reflective of all replicates (values reported as mean plus variance, including tests of significance).

Specific major concerns essential to be addressed to support the conclusions:

1) The data shows that IFITM3 blocks viral entry and that IFITM3 G95L is impaired in influenza A virus entry. However, additional experimental data are needed to demonstrate that IFITM3 modulates membrane fusion of endosomal compartments with lysosomal compartments. For this authors should take advantage of well-established early and endosomal (Rab5 and Rab7) and lysosomal markers (LAMP) fused to fluorescent proteins and measure their colocalization and organelle size distribution in HEK stable cell lines expressing IFITM3, IFITM3 G95L and also in HeLa WT and HeLa IFITM3 KO. This should confirm whether IFITM3 is regulating homotypic endosomal fusion. In the presence of IFITM3, lysosomal compartments but not endosomal compartments should be larger.

We thank the reviewer for this excellent suggestion. As instructed, we performed imaging of Rab5-RFP/Rab7-mCherry and LAMP1-GFP in 293T cells to address the reviewer's point on the distribution of early endosomes, late endosomes, and lysosomes and the extent to which they fuse with one another. We found that the colocalization between Rab7-mCherry and LAMP1-GFP was enhanced in cells expressing IFITM3 WT compared to cells expressing Empty Vector or IFITM3 G95L (Appendix Figure S2). Furthermore, as the reviewer predicted, in cells expressing IFITM3 WT, the size of the LAMP1-GFP+ compartment was expanded compared to Empty Vector and IFITM3 G95L (Appendix Figure S2 and S3). The enlarged LAMP1+ compartment that also colocalizes significantly with Rab7 is consistent with observations made in a previous publication (PMID: 22046135). In contrast, we did not see much colocalization

between Rab5 and LAMP1 in any conditions. Together, these observations provide further support that IFITM3 promotes fusion between late endosomes and lysosomes.

2) Colocalization of IFITM3 and IFITM3 G95L with SNX7 should be provided to support the results of immunoprecipitation experiments. The authors mention in line: 282 "To test the functional association between SNARE modulation and lysosomal targeting ..." However, this was not directly tested in Figure 5.

We followed the advice of the reviewer and transfected cells expressing IFITM3 WT or IFITM3 G95L with STX7-GFP and measured its colocalization with IFITM3 (identified with anti-FLAG staining). In agreement with our co-immunoprecipitation experiments, we found that IFITM3 WT colocalized with STX7-GFP to a greater extent than IFITM3 G95L did. These results are now found in new Figure 3B.

3) To prove that the IFITM3 enhances virus degradation in lysosomes, authors should provide quantitative data on lysosome cargo degradation. They could use established published protocols: Protocol for quantification of the lysosomal degradation of extracellular proteins into mammalian cells (10.1016/j.xpro.2021.100975). In addition, lysosomal degradation of EGFR could be compared in the presence of IFITM3 and IFITM3 G95L as mentioned on Line 397. One would expect that IFITM3 but not IFITM3 G95L will increase cargo degradation.

We thank the reviewer for this suggestion. We decided to address this request using two distinct yet complementary assays. First, we performed an *in vivo* late endosome-lysosome fusion assay in living cells which makes use of Dextran as endocytic cargo and is a standard in the endocytic trafficking field. We pulsed cells with Dextran and imaged its delivery to lysosomes three hours later, in living cells. Active lysosomes were labeled with Magic Red, a substrate for proteolytically active cathepsin enzymes. The Dextran/Magic Red approach directly addresses the question of whether IFITM3 regulates late endosome-lysosome fusion. Our results indicate that IFITM3 WT, but to a lesser extent G95L, promotes the fusion of Dextran+ late endosomes with Magic Red+ lysosomes, providing the direct evidence that IFITM3 promotes late endosome-lysosome fusion. These data are now found in new Figure 6B. Additionally, we found that Dextran colocalized to a greater extent with Magic Red in HeLa WT cells compared to HeLa IFITM3 KO cells, indicating that endogenous IFITM3 promotes late endosome-lysosome fusion as well (Appendix Figure S1B).

The second approach was to measure the degradation of EGFR in cells expressing IFITM3 WT or IFITM3 G95L. To do so, we utilized an EGFR-GFP construct and tracked its abundance in cells following EGF addition using flow cytometry, an assay outlined in publication PMID: 26101353. We found that EGFR-GFP degradation was slower in cells expressing IFITM3 G95L compared to IFITM3 WT, suggesting that IFITM3 WT accelerates the turnover of EGFR in a G95-dependent manner. We also introduced the EGFR-GFP into HeLa WT and HeLa IFITM3 KO and we found that EGFR-GFP loss was accelerated in WT cells compared to IFITM3 KO cells. These new data can be found in Appendix Figure S4A and S4B.

4) In addition, a control colocalization analysis of IFITM3 and IFITM3 G95L with endosomal, and lysosomal markers should be done to ensure that IFITM3 G95L targeting into endosomes is not impaired.

We agree that it is important to establish whether the G95L mutation impacts endosomal targeting of IFITM3, because that may be one reason why it exhibits decreased activity. However, we have already measured endosomal localization of IFITM3 WT and G95L in a previous publication (PMID: 33112230). We found that G95L did not significantly impact the localization of IFITM3 to early and late endosomes (as identified by EEA1 and CD63,

respectively). In other words, we found that IFITM3 WT was localized to EEA1+ and CD63+ compartments, as well as the cell surface, and the distribution of G95L to these sites was similar. These results suggest that the targeting of G95L to endosomes is not impaired. Based on these published findings, we did not think it was absolutely necessary to perform additional experiments \ to compare the subcellular localization of IFITM3 WT compared to IFITM3 G95L.

5) Most of the conclusions are based on WB analysis of bands. Even though the authors mention that WB was repeated three times, they report a single value below the WB and it is not explained if this is an average of 3 repetitions or just determined from a single WB. Are these biological or technical replicates using the same samples? How many times were the WB repeated and what is the standard deviation of the data? In some WB there is not only one band but two bands are visible and the second band is cropped away. Why not including the second band in the analysis? Uncropped WB images for all experiments and replications should be provided.

In the revised figures, we now present western blot quantifications that are reflective of multiple experiments (mean plus standard error for 3 biological replicates). The mean, variance, and statistical significance are now indicated directly in the figures when quantitation was performed. We fixed the blots where only one of two bands was visible (namely, the STX7-HA protein in whole cell lysates). As part of the Source Data, we will of course provide uncropped blot images corresponding to the images found in the main and Extended View figures, as is mandated by EMBO Journal. We will carefully heed the instructions of staff in this regard. Thank you.

6) Discussion about the structure of the amphipathic helix (Line 361-367): IFITM3 Cys72 is palmitoylated and Cys72 palmitoylation was shown to be important for IFITM3 function. It is localized between 2 amphipathic helices which are modelled here as one helix. Was palmitoylation at Cys72 used in the AlphaFold3 model presented in Figure 2C? Considering that palmitoyl is embedded in the membrane, would the SNARE complex formation with IFITM3 be sterically possible?

The reviewer brings up an interesting point. Palmitoylation of cysteine residues has been previously shown to be important for the antiviral activity of IFITM3. However, our Alphafold analysis did not take into account lipid modifications, including palmitoylation, when predicting structural folds. We believe that one or more palmitoylation sites confer membrane binding/anchoring to IFITM3, and membrane interactions are likely to be an important aspect of how IFITM3 engages with SNARE machinery and regulates SNARE assembly. Alternatively, palmitoylation of IFITM3 may be important for its intracellular trafficking, and as such, may impact which SNARE proteins that IFITM3 encounters in the cell. We plan to explore these possibilities experimentally in future work in order to directly test whether membrane modifications contribute to regulation of SNARE-mediated fusion. Of note, several genuine SNARE proteins, including STX7 and STX8, are known to be palmitoylated as well. Therefore, the presence of palmitoylation on IFITM3 does not conflict with its function as a regulator of SNARE activity. Rather, palmitoylation is yet another commonality shared between IFITM3 and genuine SNARE proteins.

7) IFITM3 is a transmembrane protein, gel filtration chromatography elution profile should be provided to show that recombinant IFITM3 was not aggregated after affinity purification. We thank the reviewer for this excellent suggestion. To rule out that our recombinant proteins were prone to aggregation, we performed size exclusion chromatography on our recombinant IFITM3 proteins (WT, F75/78A, and G95L) and subjected the eluted fractions to SDS-PAGE and

Coomassie blue staining. The results show that WT, F75/78A, and G95L protein eluted in similar fractions (namely fractions 7 through 12, which correspond to volumes of 7 mL and 12 mL, respectively). IFITM3 WT was also found in fractions 14, 15, and 16. In an independent run, we used Blue Dextran to identify that fraction 6 (6 mL) is the “void” fraction that contains aggregates. Therefore, if protein aggregates were present, we would expect to observe them eluted in fraction 6. Since our recombinant proteins eluted at fractions 7 through 12, primarily, we conclude that none of our recombinant proteins are made up of predominantly large aggregates. While it is possible that a fraction of each recombinant protein exists as aggregated protein, their elution patterns suggest that the majority of protein is not aggregated. Furthermore, since the elution patterns for the three proteins (WT, F75/78A, and G95L) were similar, a tendency for one or more of our proteins to aggregate cannot explain the unique ability of IFITM3 WT to pull down STX7. It is also important to keep in mind that, in order to perform SEC and evaluate the protein eluates in this manner, we were required to use protein inputs that far exceeded those used in our in vitro coimmunoprecipitations. Therefore, we conclude that it is very unlikely that the results from our in vitro pull down experiments are confounded or biased by protein aggregation. These data can now be found in new Extended View Figure 5A. At line 243, we state “With size exclusion chromatography, we ruled out that the selective pull down of IFITM3 WT by recombinant STX7 was due to a tendency for IFITM3 WT to form aggregates in vitro (Extended View Figure 5A).”

Minor concerns that should be addressed:

- The abstract should be restructured and shortened, the focus of the study is not immediately clear from the abstract.

The abstract has been modified and shortened as requested. Thank you.

- IFITM3 should be better introduced in the introduction including more details on the mechanism of action

At line 71 of the Introduction, we included some additional sentences to better introduce IFITM3 in the introduction (this was text moved from the Results section, where this information was found initially).

- Line 222: Please clarify what "HeLa cells competent for IFITM3" mean. Were these cells overexpressing IFITM3 or was IFITM3 induced by interferon treatment?

We have replaced “HeLa cells competent for IFITM3” with “HeLa WT cells.” We apologize for the confusion.

- Based on the proposed mechanism, how would IFITM1 localized in plasma membrane function or does interaction with SNARE complex only apply to IFITM3 and IFITM2? For example, a previous study (Kun Li et al, 2013) showed that IFITM1 can arrest cell-cell fusion.

Since the R-SNARE-like motif is conserved in human IFITM1, IFITM2, and IFITM3, we speculate that all three proteins are endowed with SNARE-binding ability. As the reviewer suggests, it would be interesting to determine whether the differential subcellular localization of IFITM1, IFITM2, and IFITM3 allows the proteins to interact with different SNARE proteins. It is possible that different IFITM proteins regulate different SNARE-mediated fusion processes, and this will be the subject of our future work.

- Overall the manuscript would be improved by including a model as a figure on the proposed molecular mechanism

We really like this suggestion by the reviewer. We now include a summary model figure at the end of the paper (new Figure 8). We believe this represents a nice addition to the paper and helps

summarize our findings to readers, making it easier for the significance to be appreciated.

- Terms like "fresh insight", "newfound", and "leap" should be avoided.

As requested, we have purged these words from the manuscript. Thank you.

Referee #3:

Rahman et al. show that members of the CD225 protein family harbor a putative R-SNARE motif, which interacts with Q-SNAREs and modulates SNARE complex assembly. Importantly, the work shows that mutations within this SNARE-like motif result in reduced SNARE binding and affect protein function. Many of these mutations have been previously associated with respective diseases. Beside some basic characterization of the putative SNARE motifs of PRRT2 and TUSC5, an in depth analysis of IFITM3 indicates that its SNARE-like motif contributes to the redirection of intracellular virus trafficking towards lysosomal degradation. Overall, the identification of SNARE motif-like sequences within the CD225 protein family profoundly forwards our understanding of the regulation of membrane trafficking and suggests that distinct CD225 family members may regulate different SNARE interactions. The data are overall compelling and the paper is well written.

We thank the reviewer for their thorough and positive assessment of our work, and for understanding its significance to SNARE biology and to our understanding of the functions played by the CD225 protein family.

Nevertheless, the manuscript could be improved by providing more detailed insights into the properties (SNARE-specific interactions) of the putative R-SNARE motif of IFITM3. The authors suggest that the IFITM3 mainly binds the Q-SNARE STX7. Indeed, ectopic protein expression in HEK293 cells (Fig. 2C and supplemental Fig. 2C) and purified proteins (Fig. 3C) confirm an IFITM3 - STX7 interaction. However, instead of using a Western blot analysis (Fig. 3C), the authors should consider to repeat the pull downs with increasing concentrations of the IFITM3 variants followed by an analysis using SDS-PAGE and Coomassie blue staining (incl. quantification).

We thank the reviewer for this suggestion. We considered assessing co-immunoprecipitation of recombinant proteins by using SDS-PAGE and Coomassie, but this was not possible due to the low levels of recombinant proteins used in the co-immunoprecipitation. We decided to use Western blotting to assess co-immunoprecipitation as this rendered the assay much more sensitive. Rather than increase the levels of recombinant proteins and assess co-immunoprecipitation by SDS-PAGE and Coomassie, which could increase the likelihood of protein aggregation and confound our measurements of protein-protein interactions, we decided that the Western blotting approach using low levels of recombinant proteins was more reliable and less subject to artifactual bias.

(Technically, the authors could also use the protein purification tag for the pull down instead of the anti-STX7 antibody.)

We used the anti-STX7 antibody as opposed to an antibody raised against the purification tag because anti-STX7 should be capable of detecting all forms of recombinant STX7, directly, in the reaction mixture and co-IP. In contrast, if the purification tag is lost, even in a portion of the recombinant protein, our detection of recombinant STX7 would be underestimated. Based on the

fact that the anti-STX7 antibody yielded a clear, highly resolved band, we were content with the results generated this way.

In case of saturable binding, such an experiment will provide a rough estimate about the binding affinities of IFITM3 variants.

We chose not to provide an estimate of the exact binding affinities. We believe what was most important was to measure the relative binding affinities of IFITM3 WT compared to IFITM3 mutants. Since we observed no detectable interaction between recombinant STX7 and IFITM3 F75/78A or IFITM3 G95L, this approach allowed us to conclude that IFITM3 WT exhibits a relatively higher potential for binding to STX7.

If possible, such binding experiments should also be performed with recombinant VAMP7 and VAMP8 (e.g. containing an affinity tag). Please note that CD225 family members also show direct v-SNARE interactions. PRRT2 directly interacts with VAMP2 (Coleman et al., 2018, Cell Reports) and a recent screen for IFITM3 binding partners identified VAMP7 as a potential partner (Li et al., 2024, International Journal of Biological Macromolecules). Importantly, potential differential v-SNARE - IFITM3 interactions may also provide mechanistic insights of how IFITM3 redirects viral trafficking. The cellular pull down results shown in Fig. 2D are not conclusive for VAMP7 and VAMP8 due to vastly different expression levels. (Reverse co-immunoprecipitation experiments were not performed.) Actually, the data shown in supplemental Fig. 4A and Fig. 4C would support a positive effect of IFITM3 for VAMP7 interactions.

We thank the reviewer for suggesting that we closely examine the binding potential between IFITM3 and VAMPs, given the prior literature on PRRT2 and some clues suggesting that IFITM3 may interact with VAMP7. However, rather than addressing this question using recombinant VAMP proteins (which would require us to confirm their purity and exclude that they are prone to aggregation), we ectopically expressed IFITM3 (WT or G95L) along with VAMP8-HA or VAMP7-Myc in cells and performed co-immunoprecipitation. Our results indicate that IFITM3 WT pulls down ectopic VAMP8 and VAMP7 under these conditions, while the G95L mutation results in less pull down (new Extended View Figure 3A and 3B).

Furthermore, we addressed whether endogenous IFITM3 was capable of interacting with endogenous VAMP7 and endogenous VAMP8 in HeLa cells, which more accurately captures the physiologically relevant cellular environment. We now provide results showing that endogenous IFITM3 pulls down endogenous VAMP7 (confirming the result seen in Li et al.) as well as endogenous VAMP8 (new Figure 4D and 4E). These results are discussed starting at line 217.

These additional results may be important for our ongoing work to determine the precise mechanism by which IFITM3 inhibits assembly of the SNARE assemblies driving homotypic late endosome fusion. Does IFITM3 sequester VAMP8 and prevent its inclusion in the complex? Does IFITM3 and its affinity for the Q-SNAREs STX7, STX8, and Vti1b allow it to swap places with VAMP8 in the complex? Does IFITM3 recruit VAMP7 to the complex? Do all of these mechanisms contribute to the effect of IFITM3 on homotypic late endosome fusion? We plan to address these possibilities and hope that others in the field do so, too.

Additional points:

- Most of the Western blot analysis, showing protein interacts, contain quantifications. It seems that these pull down experiments were performed three times. This should be reflected in the quantifications, better illustrating reproducibility.

We have now updated the mean quantifications in all figures, and they are now reflective of multiple experiments. The results are shown as mean and standard error and significant differences are indicated. We thank the reviewer for this suggestion.

• Figure 1B: To firmly exclude that different expression levels of the PRRT2 variants cause the reduced binding of the PRRT2 mutants in the pull down with SNAP-25-HA, the authors should consider to show a shorter exposure of anti-FLAG signal in the whole lysate. (As a note, in supplemental Fig. 1, all PRRT2 constructs show similar expressions levels and the mutants do not affect STX1A binding.) In this context, please also show a shorter exposure of the TUSC5 levels in the whole lysate (supplemental Fig. 1C). Can the authors provide some information about the pull down efficiencies?

As indicated by the reviewer, we noticed that the relative expression levels of PRRT2 WT compared to mutants were different when they were co-expressed with SNAP-25, but this was not the case when they were co-expressed with STX1A. For the reviewer's sake, here are the requested shorter exposures of the blots (specifically, the shorter exposure is shown for PRRT2-FLAG on the left and TUSC5-FLAG on the right):

In these examples, it is suggestive that mutations in the R-SNARE-like motif of PRRT2 and TUSC5 may result in somewhat decreased protein expression compared to WT. However, we don't think that differential protein expression fully explains the differential binding with SNAP25 or STX4. However, just to confirm our findings, we performed the reverse co-immunoprecipitation to confirm TUSC5/STX4 binding and its determinants. This reverse co-IP confirmed our initial findings—the G141W mutation in TUSC5 abrogates binding to STX4 (new Extended View Figure 1D). The results suggest that the G141W mutation in TUSC5 causes loss of STX4 binding in a manner that cannot be explained by differential protein expression. We do not have any information regarding pull down efficiencies. As mentioned in a previous point, we chose to assess relative efficiencies of pull down (rather than absolute, exact efficiencies) because our conclusions are limited to how mutations in the R-SNARE-like motif of CD225 proteins affect SNARE pull down. In that sense, we aren't attempting to address how IFITM3, PRRT2, or TUSC5 exhibit a particular affinity or pull down efficiency of SNAREs. Instead, we focus on the fact that WT versions of these proteins display binding to SNAREs by co-IP, and relatively less binding occurs when they are mutated in their R-SNARE-like motifs.

• In Figure 2C, assuming that IFITM3 indeed functions as an R-SNARE, the authors may consider to include a potential model of STX7/STX8/Vti1b/IFITM3 complex, also showing the positions of the tested mutants and how these mutants may affect/interfere with IFITM3-SNARE complex formation/interaction.

We thank the reviewer for this suggestion. We aren't making the assumption that IFITM3 is capable of acting as a genuine R-SNARE—this will require further experimental work, which we are preparing for as part of a future study. However, we added additional images of the R-SNARE-like motif of IFITM3 predicted by AlphaFold in order to indicate the positions of key residues F75, F78, and R85. Here, we modeled the R-SNARE-like motif of IFITM3 together with the Q-SNARE motifs of STX7, STX8, and Vti1b, and labeled the F75, F78, and R85

residues of IFITM3 to visualize how they may interact with the Q-SNAREs. Importantly, the side chains of all three residues face inwards towards the core of the coiled-coil structure formed by the alpha-helices of IFITM3, STX7, STX8, and Vti1b. Moreover, we zoomed in on the R85 residue of IFITM3 and found that its side chain is positioned towards the Q residues of STX7, STX8, and Vti1b, which is structurally similar to how the central R of VAMP8 interacts with the Q residues of these same SNARE proteins. These results can now be found in new Extended View Figure 2B and 2C. These findings provide a structural explanation for why mutations at F75, F78, and R85 of IFITM3 result in loss of SNARE binding activity (and loss of antiviral activity).

- In supplemental Fig. 2C, the F75/78A mutant, compared to the IFITM3 WT, shows enhanced binding to Vti1b (despite a lower expression level). Please provide an explanation.

We also noticed that the F75/78A mutant of IFITM3 shows enhanced binding to Vti1b, but less binding to STX7 and STX8, relative to WT. This discrepancy is surprising, and we do not have an explanation at the time. Of note, IFITM3 pulled down with STX7 and STX8 to a higher extent than Vti1b, so it is unclear whether this “enhanced” binding is functionally significant. The results may suggest that IFITM3 can interact preferentially with certain Q-SNAREs compared to others, but also, that the interactions with different Q-SNAREs involve slightly different regions of the R-SNARE-like motif of IFITM3. It is possible that the interaction between IFITM3 and a given Q-SNARE may require a prior interaction with a different Q-SNARE.

- In Fig. 3B, supplemental Fig. 3B, and Fig 4B, it seems that the isotypic antibody precipitate STX8. Please clarify.

The band which is present in the isotype antibody IP lane is light chain Ig, which is very close in size to STX8 (~25 kD). The anti-STX8 antibody clearly immunoprecipitates STX8, because the immunoblotting with anti-STX8 antibody reveals signal that is more intense than the light chain Ig. This is not problematic for our conclusions, since the light chain Ig is present and equal in all lanes of the IP. We have now marked the light chain Ig with a # symbol and we describe it as a marker for the light chain Ig in the figure captions. These results can now be found in new Figure 4B and new Extended View Figure 4B).

- In Materials and Methods, it is mentioned that supplemental Fig. 3D shows Coomassie blue staining, but the corresponding figure legends mentions immunoblotting with anti IFITM3 antibody. Please clarify. Why do the authors show an immunoblot, instead of a Coomassie blue stain of purified STX7 in supplemental Fig. 3E?

We apologize for the confusion. We fixed the figure labeling and now show Coomassie staining of recombinant IFITM3 proteins in new Extended View Figure 4C and immunoblotting in Extended View Figure 4D. The text has been corrected to reflect this change. Also, we added Coomassie blue staining for the recombinant STX7, and this can now be found in Extended View Figure 4E. We thank the reviewer for pointing out this omission.

- In the Western blot analysis shown in Fig. 4D and supplemental Fig. 4B please also show the expression levels of the ectopic SNAREs (protein bands corresponding to MW of monomeric state). Additional Western blot analysis of the SDS-resistant bands (Fig. 4D) using anti-VAMP7 and anti-IFITM3 (FLAG) antibodies could further strengthen the manuscript. (For example, IFITM3 may not be part of the SDS-resistant complex, but VAMP7 may be increased in the presence of IFITM3.)

We thank the reviewer for this suggestion. We have included the bands corresponding to SNARE monomers in the relevant blots, which are now found in new Figure 6A (anti-Myc (VAMP8)

immunoblot) and Appendix Figure S1A (anti-STX8 immunoblot). We only probed for VAMP8 and STX8, but did not probe for the other SNARE proteins (STX7, Vti1b), since they contained an HA tag in common. We attempted to detect VAMP7 in the SDS-resistant complexes, but we did not observe detectable signal in any of the conditions tested (Empty Vector, IFITM3 WT, IFITM3 G95L). We also probed for IFITM3 with anti-FLAG and could only visualize IFITM3 in its monomeric state (it was not detected in the SDS-resistant complex). The lack of detection of IFITM3 in a higher-order, SNARE-containing complex is one reason why we are not yet claiming that IFITM3 acts as a genuine R-SNARE.

- Supplemental Fig. 4, figure legend: please replace the typo "IIP" by "IP".

We have corrected this mistake. Thank you for bringing it to our attention.

- In most Western blot analysis, the Ig chain seems to migrate at 35 kDa. In others, it migrates above 40 kDa (e.g. Fig. 3A and B). The IgG heavy and light chains usually migrate at 50 and 25 kDa, respectively. Please briefly explain the discrepancy.

We apologize for the confusion. We have now modified the labeling of all immunoprecipitation data to precisely indicate whether the heavy or light Ig chains are shown. We have also updated the tick marks to indicate the MW of the heavy or light Ig chains (approximately 50 and 25 kD, respectively). Thank you.

Dear Alex,

Thank you for submitting a revised version of your manuscript. We have now received input from all original reviewers, who find that their previous concerns have been addressed satisfactorily and recommend acceptance of the manuscript after minor revisions requested in the comments by reviewers #1 and #3.

Additionally, there remain a few editorial points that need addressing before I can extend official acceptance of the manuscript:

1. Please check if the email addresses provided for the authors Abigail Jolley (abigail.jolley@nih.gov), Kazi Rahman (kazi.rahman@nih.gov) are correct, as the emails sent to them were returned.
2. Please submit up to five keywords.
3. Please add a heading "Expanded View Figure Legends" after main figure legends and update the nomenclature for Expanded View figures to that of figure EV1-EV5 in the manuscript text, the figure titles and callouts.
4. Please rename "Declaration of interests" section into "Disclosure and competing interests statement" (further info: <https://www.embopress.org/page/journal/14602075/authorguide#conflictsofinterest>).
5. Please update references according to The EMBO Journal style - where there are more than 10 authors on a paper, the first 10 should be listed, followed by 'et al.' Please see further information here: <https://www.embopress.org/page/journal/14602075/authorguide#referencesformat>
6. There is a reference to "data not shown" on page 6, line 233. Our policy does not permit references to "data not shown", we usually ask to include this information in the Appendix. In this case, since it not particularly informative to provide an empty blot, please simply remove this reference.
7. Please remove the "Reagent availability" statement from the Data availability section.
8. Please check the order of the figure callouts: currently, Figure 3A is called out before Figure 2E.
9. In the Appendix, please add page numbers in the table of contents. Please move the legends underneath the corresponding figure, as the appendix will not be typeset and will be published as provided.
10. Our data editors have flagged the following issues in figure legends that need correcting:
 - Please provide the figure titles for figures EV1, EV2, EV3, EV4, EV5.
 - Please provide the exact p values in the legends of figures 1b; 2b, e; 3a; 5a, c-d; 7b-c; EV 1c; EV 3a-b; EV 5b.
 - Please provide information on the nature and number of replicates in the legend of figure 2b.
 - Please define the error bars in the legend of figure 2b.
 - Please define the hash (#) in the legends of figures 5d, EV 4b.
11. Papers published in The EMBO Journal are accompanied online by a 'Synopsis' to enhance discoverability of the manuscript. It consists of A) a short (1-2 sentences) summary of the findings and their significance, B) 3-4 bullet points highlighting key results and C) a synopsis image that is 550x300-600 pixels large (width x height, jpeg or png format). You can either show a model or key data in the synopsis image. Please note that the image size is rather small and that text needs to be readable at the final size. Please send us this information together with the revised manuscript.

With best wishes,

Ieva

We realize that it is difficult to revise to a specific deadline. In the interest of protecting the conceptual advance provided by the work, we recommend a revision within 3 months (3rd Feb 2025). Please discuss the revision progress ahead of this time with

the editor if you require more time to complete the revisions.

Referee #1:

This work reports important findings that shed light on the antiviral activity of IFITM proteins. The authors report that, through regulating homotypic/heterotypic endosome/lysosome fusion, these proteins regulate the rate of virus degradation in lysosomes and thereby modulate the likelihood of productive fusion/infection.

In the revised manuscript, the authors addressed my prior (relatively minor) concerns. The manuscript is much improved by inclusion of live cell imaging data supporting the notion that IFITM3 inhibits homotypic fusion of late endosomes.

I have no major concerns.

Two minor points:

1. Please comment on why the Appendix Figure S1A shows a much smaller SDS-resistant complex (less than 170 kDa) than one in Figure 6A.
2. Ideally, varying the duration of cell loading with fluorescent dextran would provide a more robust assessment of endosome-lysosome fusion (colocalization with a cathepsin marker) in Figure 6B and Appendix Figure S1B.

Referee #2:

All my comments were addressed. Authors now included additional experiments supporting their conclusion that IFITM3 is able to regulate homotypic late endosomal fusion. In particular, the MagicRed Cathepsin assay provided is elegant and the provided model at the end of the manuscript nicely summarises the results. I recommend this work for publication in EMBO J.

Referee #3:

In the revised manuscript, Rahman and colleagues have added further experiments improving the manuscript and have overall addressed most of my previous concerns. Nevertheless, they should briefly resolve the following (technical) points: A technical issue, related to the previously mentioned exposure of immuno-blots: Are the authors sure that for their quantifications, the gray levels of the immune-stained bands are in the linear range? For example, in extended Fig. 5B, STX7-HA IP, it seems that saturation has been reached in the representative gel for both the HA and Myc labels, making a quantification difficult.

Extended Fig. 4C: The authors loaded 6, 12 and 18 μ g of their IFITM3 variants. Thus, WT and the variants should show similar Coomassie blue staining intensities. This is not the case, the G95L intensities seem to be at least 2 x increased compared to the WT. Furthermore, 12 μ g WT IFITM3 seems to show an increased staining intensity compared to 18 μ g sample.

Fig. 4E: The authors mention that two forms of VAMP7 were detected by immunoblotting. In case, previous publications have already described the existence of two VAMP7 forms, please mention the corresponding citation(s). Can the authors exclude the possibility that the 15 kDa band represents a cross-reactivity of the employed antibody. If appropriate, please adjust/modify the wording in the text.

Extended Fig. 4D: Just for clarity, did the authors indeed use 20 μ g of the purified antigen (IFITM3) for this immuno-blot?

Extended Fig. 5A: Please show the positions of standard gel filtration MW markers in relation to the different fractions. Actually, the SEC shows a relatively broad distribution of rIFITM3, with peaks at fractions 14,15 for WT, at fractions 7-9 for G95L, and at fractions 10,11 for F75A/78A, respectively. This indicates that the mutants show a tendency to form larger complexes, which may also affect SNARE binding.

FROM THE EDITOR:

1. Please check if the email addresses provided for the authors Abigail Jolley (abigail.jolley@nih.gov), Kazi Rahman (kazi.rahman@nih.gov) are correct, as the emails sent to them were returned.

The corrected/up-to-date email addresses for these individuals are abbie.jolley@duke.edu and rahman.kazi@northsouth.edu . Thank you.

2. Please submit up to five keywords.

membrane fusion, virus, IFITM, PRRT2, CD225

3. Please add a heading "Expanded View Figure Legends" after main figure legends and update the nomenclature for Expanded View figures to that of figure EV1-EV5 in the manuscript text, the figure titles and callouts.

We have added the heading and updated the nomenclature for the Expanded View figures as requested.

4. Please rename "Declaration of interests" section into "Disclosure and competing interests statement" (further

info: <https://www.embopress.org/page/journal/14602075/authorguide#conflictsofinterest>).

This change has been made.

5. Please update references according to The EMBO Journal style - where there are more than 10 authors on a paper, the first 10 should be listed, followed by 'et al.' Please see further information here: <https://www.embopress.org/page/journal/14602075/authorguide#referencesformat>

We have updated the citation style to the format preferred by EMBO Journal. Thank you.

6. There is a reference to "data not shown" on page 6, line 233. Our policy does not permit references to "data not shown", we usually ask to include this information in the Appendix. In this case, since it not particularly informative to provide an empty blot, please simply remove this reference.

We have removed the instance of “data not shown” at line 229 as suggested.

7. Please remove the "Reagent availability" statement from the Data availability section.

We have removed the “Reagent availability” statement at line 630.

8. Please check the order of the figure callouts: currently, Figure 3A is called out before Figure 2E.

We swapped the order of some sentences in this paragraph, starting at line 195, so that Figure 2E is now called out before Figure 3A. The text is now improved as a result. Thank you.

9. In the Appendix, please add page numbers in the table of contents. Please move the legends underneath the corresponding figure, as the appendix will not be typeset and will be published as provided.

We have remade the Appendix such that the page numbers are listed in the table of contents, and legends now appear beneath the figures. Thank you.

10. Our data editors have flagged the following issues in figure legends that need correcting:

- Please provide the figure titles for figures EV1, EV2, EV3, EV4, EV5.

Titles for the EV Figures are now provided.

- Please provide the exact p values in the legends of figures 1b; 2b, e; 3a; 5a, c-d; 7b-c; EV 1c; EV 3a-b; EV 5b.

Exact p values have now been listed in the indicated figure legends as requested. Thank you.

- Please provide information on the nature and number of replicates in the legend of figure 2b.

FROM THE REVIEWERS:

Referee #1:

This work reports important findings that shed light on the antiviral activity of IFITM proteins. The authors report that, through regulating homotypic/heterotypic endosome/lysosome fusion, these proteins regulate the rate of virus degradation in lysosomes and thereby modulate the likelihood of productive fusion/infection.

In the revised manuscript, the authors addressed my prior (relatively minor) concerns. The manuscript is much improved by inclusion of live cell imaging data supporting the notion that IFITM3 inhibits homotypic fusion of late endosomes.

We thank the reviewer for their insight and guidance during this peer review process.

I have no major concerns.

Two minor points:

1. Please comment on why the Appendix Figure S1A shows a much smaller SDS-resistant complex (less than 170 kDa) than one in Figure 6A.

Thank you for pointing out this inconsistency. The reason for why the SDS-resistant complexes in Appendix Figure S1A and Figure 6A exhibit different sizes is because we incorrectly labeled the kDa protein standards in the ladder. In both Figures, the SDS-resistant complex should appear above (larger than) the 140 kDa standard, which is the largest and final band in the ladder. This mistake was spotted by consulting the uncropped versions of the blots, which are now provided in the Source Data. In both Figures, the SDS-resistant complex migrates more slowly than the 140 kDa standard in the ladder, indicating that both complexes are larger than 140 kDa. While these gels do not allow us to calculate the exact size of the SDS-resistant complexes, what is clear is that both appear larger than 140 kDa. We have updated the figures to reflect this change. We apologize for the confusion and thank the reviewer for pointing out this error.

2. Ideally, varying the duration of cell loading with fluorescent dextran would provide a more robust assessment of endosome-lysosome fusion (colocalization with a cathepsin marker) in Figure 6B and Appendix Figure S1B.

We agree that performing the Dextran/Magic Red colocalization experiment at multiple time points post Dextran addition could have generated more insight compared to the singular time point that we assessed. However, since we observed differences between conditions at the single time point assessed, we deemed it unnecessary to perform the analysis at multiple time points. Furthermore, our flow-based EGFR degradation assay was performed at three timepoints post EGF addition (Appendix Figure S4). Therefore, we believe we covered ample ground in measuring the dynamic and progressive nature of endosome to lysosome fusion events.

Referee #2:

All my comments were addressed. Authors now included additional experiments supporting their conclusion that IFITM3 is able to regulate homotypic late endosomal fusion. In particular, the MagicRed Cathepsin assay provided is elegant and the provided model at the end of the manuscript nicely summarises the results. I recommend this work for publication in EMBO J. We thank the reviewer for thoroughly assessing our manuscript and for their critical and productive comments that led to a significantly improved final product.

Referee #3:

In the revised manuscript, Rahman and colleagues have added further experiments improving the manuscript and have overall addressed most of my previous concerns.

We thank the reviewer for their constructive feedback that led to significant improvements of our manuscript.

Nevertheless, they should briefly resolve the following (technical) points:

A technical issue, related to the previously mentioned exposure of immuno-blots: Are the authors sure that for their quantifications, the gray levels of the immune-stained bands are in the linear range? For example, in extended Fig. 5B, STX7-HA IP, it seems that saturation has been reached in the representative gel for both the HA and Myc labels, making a quantification difficult.

We are sure that saturation was not reached in this or any of the other immunoblots found in our manuscript, because the Li-COR ImageStudio software used for analysis includes an automated quality control step whereby saturated pixels are detected and flagged. This process introduces pseudocolored pixels into the image to highlight points of saturation. No such instances of saturation were detected in the blots appearing in this manuscript. Thank you.

Extended Fig. 4C: The authors loaded 6, 12 and 18 μg of their IFITM3 variants. Thus, WT and the variants should show similar Coomassie blue staining intensities. This is not the case, the G95L intensities seem to be at least 2 x increased compared to the WT. Furthermore, 12 μg WT IFITM3 seems to show an increased staining intensity compared to 18 μg sample.

We recognize that the Coomassie blue staining intensities are not identical between the recombinant IFITM3 WT and mutants. However, based on our previous experiences, we do not use Coomassie blue staining to quantify protein amounts, nor do we depend upon Coomassie blue for confirming the relative amounts of protein loaded into the gel. Rather, we measure protein concentrations directly by UV absorbance at 280 nm and by the Bradford assay. We are unsure why the staining intensity of 12 micrograms of protein can sometimes appear more intense than 18 micrograms of the same sample, but we consider it a limitation of the Coomassie blue staining technique. The purpose of Figure EV4C was to assess the purity and apparent size of the three recombinant proteins, not their concentrations relative to one another. The fact that three different concentrations of protein were loaded into the gel was done to ensure we would be able to clearly detect each protein by this method.

Fig. 4E: The authors mention that two forms of VAMP7 were detected by immunoblotting. In case, previous publications have already described the existence of two VAMP7 forms, please mention the corresponding citation(s). Can the authors exclude the possibility that the 15 kDa band represents a cross-reactivity of the employed antibody. If appropriate, please adjust/modify the wording in the text.

We believe our immunoblotting of VAMP7 resulted in the detection of two forms of VAMP7 of differing sizes, because we consulted this reference from a leading lab in the field of SNARE biology (particularly VAMP7): Wojnacki et al., Cell Reports, 2020 PMID: 33357422. In this paper, they state “we detected the full-length VAMP7 and a 15-kDa fragment likely corresponding to the cytoplasmic domain of VAMP7 as our antibody was generated against this domain.” Moreover, we do not think this is a critical point because, after all, the anti-VAMP7 antibody was able to pull down IFITM3 in WT but not IFITM3 KO cells, and that is the point of the figure. The 15 kDa band detected by anti-VAMP7 is not IFITM3, since IFITM3 migrates slightly more quickly (apparent size less than 15 kDa, as determined by anti-IFITM3 used on the same membrane), and the 15 kDa band detected by anti-VAMP7 is also present in IFITM3 KO cells. We added the reference into the Figure 4 figure legend.

Extended Fig. 4D: Just for clarity, did the authors indeed use 20 μg of the purified antigen (IFITM3) for this immuno-blot?

We thank the reviewer for pointing out this mistake. The text should have indicated that 20 nanograms of recombinant IFITM3 was examined by immunoblotting. This has been corrected in the Figure EV4 figure legend.

Extended Fig. 5A: Please show the positions of standard gel filtration MW markers in relation to the different fractions. Actually, the SEC shows a relatively broad distribution of rIFITM3, with peaks at fractions 14,15 for WT, at fractions 7-9 for G95L, and at fractions 10,11 for F75A/78A, respectively. This indicates that the mutants show a tendency to form larger complexes, which may also affect SNARE binding.

As requested, we examined the elution profiles of two proteins commonly used as markers during SEC, bovine serum albumin and RNase A, by measuring absorbance at 280 nm throughout the elution volumes. The chromatogram is shown below. We agree with the reviewer that recombinant G95L and F75/78A may have a slightly higher tendency to form large complexes compared to WT. However, we think that if G95L and F75/78A showed a significantly higher tendency to form larger complexes, this would increase the likelihood that they would pull down with recombinant syntaxin 7 (but we did not observe that –we only detected co-immunoprecipitation between IFITM3 WT and syntaxin 7).

Dear Alex,

Thank you for addressing the final editorial points. I would like to apologise again for the delay in the handling of your manuscript due to conference travel and the resulting backlog. I am now pleased to inform you that your manuscript has been accepted for publication in the EMBO Journal.

I am currently working on the final textual editing of the synopsis and abstract and will share any changes, if such will be needed, at the beginning of the next week.

If you have any questions, please do not hesitate to contact the Editorial Office. Thank you for this contribution to The EMBO Journal and congratulations on a nice study!

Best wishes,

Ieva

Ieva Gailite, PhD
Senior Scientific Editor
The EMBO Journal
Meyerohofstrasse 1
D-69117 Heidelberg
Tel: +4962218891309
i.gailite@embojournal.org
